# A Unifying View of Optimism in Episodic Reinforcement Learning

**Gergely Neu**
Universitat Pompeu Fabra
Barcelona, Spain
gergely.neu@gmail.com

**Ciara Pike-Burke**[*]
Imperial College London
London, UK
c.pikeburke@gmail.com

## Abstract

The principle of "optimism in the face of uncertainty" underpins many theoretically successful reinforcement learning algorithms. In this paper we provide a general framework for designing, analyzing and implementing such algorithms in the episodic reinforcement learning problem. This framework is built upon Lagrangian duality, and demonstrates that every *model-optimistic* algorithm that constructs an optimistic MDP has an equivalent representation as a *value-optimistic* dynamic programming algorithm. Typically, it was thought that these two classes of algorithms were distinct, with model-optimistic algorithms benefiting from a cleaner probabilistic analysis while value-optimistic algorithms are easier to implement and thus more practical. With the framework developed in this paper, we show that it is possible to get the best of both worlds by providing a class of algorithms which have a computationally efficient dynamic-programming implementation and also a simple probabilistic analysis. Besides being able to capture many existing algorithms in the tabular setting, our framework can also address large-scale problems under realizable function approximation, where it enables a simple model-based analysis of some recently proposed methods.

## 1 Introduction

Reinforcement learning (RL) is a key framework for sequential decision-making under uncertainty [45, 46]. In an RL problem, a learning agent interacts with a reactive environment by taking a series of actions. Each action provides the agent with some reward, but also takes them to a new state which determines their future rewards. The aim of the agent is to pick actions to maximize their total reward in the long run. The learning problem is typically modeled by a Markov Decision Process (MDP, [40]) where the agent does not know the rewards or transition probabilities. Dealing with this lack of knowledge is a crucial challenge in reinforcement learning: the agent must maximize their rewards *while simultaneously* learning about the environment. One class of algorithms that have been successful at balancing this *exploration versus exploitation* trade-off are *optimistic reinforcement learning algorithms*. In this paper, we provide a new framework for studying this class of algorithms.

Optimistic algorithms are built upon the principle of "optimism in the face of uncertainty" (OFU). They operate by maintaining a set of statistically plausible models of the world, and selecting actions to maximize the returns in the best plausible world. Such algorithms were first studied in the context of multi-armed bandit problems [29, 2, 14, 5, 30], and went on to inspire numerous algorithms for reinforcement learning. A closer look at the literature reveals two main approaches to incorporate optimism into RL. In the first, optimism is introduced through estimates of the MDP: these approaches build a set of plausible MDPs by constructing confidence bounds around the empirical transition and

---

[*]This work was done while CPB was at Universitat Pompeu Fabra and Barcelona Graduate School of Economics.

reward functions, and select the policy that generates the highest total expected reward in the best feasible MDP. We refer to this family of methods as *model-optimistic*. Examples of model-optimistic methods include RMAX [13, 27, 47] and UCRL2 [4, 24, 44]. While conceptually appealing, model-optimistic methods tend to be difficult to implement due to the complexity of jointly optimizing over models and policies. Another approach to incorporating optimism into RL is to construct optimistic upper bounds on the *optimal value functions* which are (informally) the total expected reward of the optimal policy in the true MDP. The optimistic policy greedily picks actions to maximize the optimistic values. We refer to this class of methods as *value-optimistic*. Examples of algorithms in this class are MBIE-EB [44], UCB-VI [6] and UBEV [16]. These algorithms compute the optimistic value functions via dynamic programming (cf. 9), making them computationally efficient and compatible with empirically successful RL algorithms that are typically based on value functions. One downside of these approaches is that their probabilistic analysis is often excessively complex, with complicated recursive arguments necessary to guarantee optimism.

While these two approaches may look very different on the surface, we show in this paper that there is in fact a very strong connection between them. Our first contribution is to show that the optimization problems associated with these two approaches exhibit strong duality. This implies that that for every model-optimistic approach, there exists an equivalent value-optimistic approach. This bridges the gap between the conceptually simple model-optimistic approaches and the computationally efficient value-optimistic approaches. This result enables us to develop a general framework for designing, analyzing and implementing optimistic algorithms in the episodic reinforcement learning problem. Our framework is broad enough to capture many existing algorithms for tabular MDPs, and for these we provide a simple analysis and computationally efficient implementation. The framework can also be extended to incorporate realizable linear function approximation, where it leads to a new model-based analysis of two value-optimistic algorithms. Our analysis involves constructing a new model-optimistic formulation for factored linear MDPs which may be of independent interest.

## 2   Background on Markov Decision Processes

**Finite-horizon episodic MDPs.**   A finite episodic Markov decision process (MDP) is a tuple $(\mathcal{S}, \mathcal{A}, H, \alpha, P, r)$ where $\mathcal{S}$ and $\mathcal{A}$ are the finite sets of states and actions with $S = |\mathcal{S}|, A = |\mathcal{A}|$, $H$ is the (fixed) episode length and $\alpha$ is the initial state distribution. The transition functions, $P = \{P_h(\cdot|x,a)\}_{h,x,a}$, give the probability $P_h(x'|x,a)$ of reaching state $x' \in \mathcal{S}$ after playing action $a \in \mathcal{A}$ from state $x \in \mathcal{S}$ at stage $h$ of an episode, and the reward function, $r : \mathcal{S} \times \mathcal{A} \to [0,1]$, assigns a reward to each state-action pair. For simplicity, we assume $r$ is known and deterministic[2], and each episode $t$ begins from state $x_{1,t} \sim \alpha$. If no further structure is assumed, we call the MDP *tabular*. We define a stationary *policy* $\pi : \mathcal{S} \to \mathcal{A}$ as a mapping from states to actions, and a nonstationary policy as a collection $\boldsymbol{\pi} = \{\pi_h\}_{h=1}^H$ of stationary policies for each stage $h$ of an episode, and note that these are sufficient for maximizing reward in an episode. We denote by $\mathbb{P}_{\boldsymbol{\pi}}[\cdot]$ and $\mathbb{E}_{\boldsymbol{\pi}}[\cdot]$ a probability or expectation with respect to the distribution of state-action sequences under policy $\boldsymbol{\pi}$ in the MDP, and also use the notations $[H] = \{1, \dots, H\}$ and $\mathcal{Z} = \mathcal{S} \times \mathcal{A}$.

**Value functions and dynamic programming.**   For any policy $\boldsymbol{\pi}$, we define the *value function* at each state $x \in \mathcal{S}$ and stage $h$ as the expected total reward from running policy $\boldsymbol{\pi}$ from that point on:

$$V_h^{\boldsymbol{\pi}}(x) = \mathbb{E}_{\boldsymbol{\pi}}\left[ \sum_{l=h}^H r_l(x_l, \pi_l(x_l)) \Big| x_h = x \right].$$

We denote by $\boldsymbol{\pi}^*$ an *optimal policy* satisfying $V_h^{\boldsymbol{\pi}^*}(x) = \max_{\boldsymbol{\pi}} V_h^{\boldsymbol{\pi}}(x)$ for all $x \in \mathcal{S}, h \in [H]$, and the *optimal value function* by $V_h^*(x) = V_h^{\boldsymbol{\pi}^*}(x)$. The total expected reward of $\boldsymbol{\pi}^*$ in an episode starting from state $x_1$ is $V_1^*(x_1)$. We define the *(optimal) action-value function* for each $x, a, h$ as

$$Q_h^{\boldsymbol{\pi}}(x,a) = \mathbb{E}_{\boldsymbol{\pi}}\left[ \sum_{l=h}^H r_l(x_l, \pi_l(x_l)) \Big| x_h = x, a_h = a \right] \qquad \text{and} \qquad Q_h^*(x,a) = \max_{\boldsymbol{\pi}} Q_h^{\boldsymbol{\pi}}(x,a).$$

It is easily shown that the value functions satisfy the *Bellman equations* for all $x, a, h$:

$$V_h^{\boldsymbol{\pi}}(x) = Q_h^{\boldsymbol{\pi}}(x, \pi(x)), \quad V_{H+1}^{\boldsymbol{\pi}}(x) = 0 \qquad\qquad V_h^*(x) = \max_{a \in \mathcal{A}} Q_h^*(x,a), \quad V_{H+1}^*(x) = 0$$

$$Q_h^{\boldsymbol{\pi}}(x,a) = r_h(x,a) + \sum_{y \in \mathcal{S}} P_h(y|x,a) V_{h+1}^{\boldsymbol{\pi}}(y) \ \text{and} \ Q_h^*(x,a) = r_h(x,a) + \sum_{y \in \mathcal{S}} P_h(y|x,a) V_{h+1}^*(y).$$

In a fixed MDP, an optimal policy can be found by solving the above system of equations by backward recursion through the stages $H, H-1, \ldots, 1$, a method known as *dynamic programming* [8, 22, 9].

**Optimal control in MDPs by linear programming.** A key technical tool underlying our results is a classic linear-programming (LP) formulation for solving MDPs [34, 18, 19]. To state this formulation, we will represent value functions by $S$-dimensional vectors and define the $S \times S$ *transition matrix* $P_{h,a}$ for each $h, a$, acting on a value function $V$ as $(P_{h,a}V)(x) = \sum_{x'} P_{h,a}(x'|x,a)V(x')$. Then, the following LP can be seen to be equivalent to the Bellman optimality equations:

$$\operatorname*{minimize}_{V} \quad V_1(x_1) \left| \begin{array}{l} \text{subject to} \\ V_h \geq r_a + P_{h,a}V_{h+1} \quad \forall a \in \mathcal{A}, h \in [H], \end{array} \right. \tag{1}$$

where the inequality is to be understood to hold entrywise. Defining the vector $q_{h,a} = (q_h(x_1,a), \ldots, q_h(x_S,a))^\mathsf{T}$, the dual of the above LP is given as

$$\operatorname*{maximize}_{q \in \mathcal{Q}(x_1)} \langle q_{h,a}, r_a \rangle \left| \begin{array}{l} \text{subject to} \\ \sum_a q_{h+1,a} = \sum_a P_{a,h}^\mathsf{T} q_{h,a} \quad \forall x \in \mathcal{S}, h \in [H], \end{array} \right. \tag{2}$$

for $\mathcal{Q}(x_1) = \{q \in \mathbb{R}_+^{S \times A \times H} : \sum_a q_1(x,a) = \mathbb{I}\{x = x_1\}, q_h(x,a) \geq 0 \ \forall (x,a) \in \mathcal{Z}, h \in [H]\}$. Feasible points of the above LP can be interpreted as *occupancy measures*. For a fixed policy $\boldsymbol{\pi}$, the occupancy measure $q^{\boldsymbol{\pi}}$ of policy $\boldsymbol{\pi}$ at the state-action pair $x, a$ is defined as $q_h^{\boldsymbol{\pi}}(x,a) = \mathbb{P}_{\boldsymbol{\pi}}[x_h = x, a_h = a]$. It can be shown that the set of occupancy measures is uniquely characterized by $\mathcal{Q}(x_1)$ and the constraint in (2). Each feasible $q$ induces a stochastic policy $\pi^q$ defined as $\pi_h^q(a|x) = \frac{q_h(x,a)}{\sum_{a' \in \mathcal{A}} q_h(x,a')}$ if the denominator is nonzero, and defined arbitrarily otherwise. The optimal solution $q^*$ to the LP in (2) can be shown to induce an optimal policy $\boldsymbol{\pi}^*$ which satisfies the Bellman optimality equations. For proofs and further details of this formulation, see Puterman [40].

**Linear function approximation in MDPs.** In most practical problems, the state space is too large to use the above results and it is common to work with parameterized estimates of the quantities of interest. We focus on the classic idea of *linear function approximation* to represent the action-value functions as linear functions of some fixed $d$-dimensional feature map $\varphi : \mathcal{S} \to \mathbb{R}^d$, so $Q_h^\theta(x,a) = \langle \theta_{h,a}, \varphi(x) \rangle$ for some $\theta_{h,a} \in \mathbb{R}^d$ for each action $a$ and stage $h$. To avoid technicalities, we assume that the state space $\mathcal{S}$ is still finite, although potentially very large. This allows us to define the $\mathcal{S} \times d$ *feature matrix* $\Phi$ with its $x^{\text{th}}$ row being $\varphi^\mathsf{T}(x)$, and represent the action-value function as $Q_{h,a} = \Phi \theta_{h,a}$. We make the following assumption:

**Assumption 1** (Factored linear MDP [51, 39, 26]). *For each action $a$ and stage $h$, there exists a $d \times \mathcal{S}$ matrix $M_{h,a}$ and a vector $\rho_a$ such that the transition matrix can be written as $P_{h,a} = \Phi M_{h,a}$, and the reward function as $r_a = \Phi \rho_a$. Furthermore, the rows of $M_{h,a}$, $m_{h,a}(x)$, satisfy $\|m_{h,a}(x)\|_1 \leq C_P$ for all $(x,a,h)$, $\rho$ satisfies $\|\rho_a\|_2 \leq C_r$, and $\|\varphi(x)\|_2 \leq R$ for some positive constants $C_P, C_r, R$.*

As shown by Jin et al. [26], this assumption implies that for every policy $\boldsymbol{\pi}$, there exists a $\theta^{\boldsymbol{\pi}}$ such that $Q_h^{\boldsymbol{\pi}}(x,a) = \langle \theta_{h,a}^{\boldsymbol{\pi}}, \varphi(x) \rangle$. We now show that factored linear MDPs also enjoy a strong dual realizability property. Let $W_{h,a}$ be an arbitrary symmetric $\mathcal{S} \times \mathcal{S}$ *weight matrix* for each action $a$ such that $\Phi^\mathsf{T} W_{h,a} \Phi$ is full rank, and notice that, due to the realizability of the action-value functions, the optimal value functions can be written as the solution to the following LP:

$$\operatorname*{minimize}_{V, \theta} \quad V_1(x_1) \left| \begin{array}{ll} \text{subject to} \\ \theta_{h,a} = (\Phi^\mathsf{T} W_a \Phi)^{-1} \Phi^\mathsf{T} W_{h,a} (r_a + P_{h,a}V_{h+1}) & \forall h \in [H], \\ V_h \geq \Phi \theta_{h,a} & \forall a \in \mathcal{A}, h \in [H]. \end{array} \right.$$

Under Assumption 1, this LP is feasible and has a finite solution. It also holds that parameter vectors $\theta_{h,a}$ are independent of the choice of the weight matrix $W_{h,a}$. The dual of this LP can be written as

$$\operatorname*{maximize}_{q \in \mathcal{Q}(x_1), \omega} \sum_{h=1}^{H} \sum_a \langle W_{h,a} \Phi \omega_{h,a}, r_a \rangle \left| \begin{array}{ll} \text{subject to} \\ \sum_a \mathbf{q}_{h+1,a} = \sum_a P_{h,a}^\mathsf{T} W_{h,a} \Phi \omega_{h,a} & \forall h \in [H] \\ \Phi^\mathsf{T} \mathbf{q}_{h,a} = \Phi^\mathsf{T} W_{h,a} \Phi \omega_{h,a} & \forall a \in \mathcal{A}, h \in [H] \end{array} \right. \tag{3}$$

Due to the boundedness and feasibility of the primal LP, the dual is also feasible and bounded. Moreover, any vector $\mathbf{q}$ that is feasible for (3) is also a feasible solution to the full LP (2), since

$$\sum_a \mathbf{q}_{h+1,a} = \sum_a P_{h,a}^\mathsf{T} \Phi W_{h,a} \omega_{h,a} = \sum_a M_{h,a} \Phi^\mathsf{T} W_{h,a} \Phi \omega_{a,h} = \sum_a M_{h,a} \Phi^\mathsf{T} \mathbf{q}_{h,a} = \sum_a P_{h,a} \mathbf{q}_{h,a}.$$

Thus, for factored linear MDPs, the set of occupancy measures is *exactly* characterized by the constraints in (3). To the best of our knowledge, these LP formulations and results are novel and may have other uses beyond the setting of factored linear MDPs. For instance, MDPs exhibiting zero inherent Bellman error [54] can be also seen to yield a feasible and finite solution for both LPs, although the above dual realizability property is not guaranteed to hold for all occupancy measures.

## 3   Regret Minimization in Episodic Reinforcement Learning

We consider algorithms that sequentially interact with a fixed but *unknown* MDP over $K$ episodes. In each episode, $t$, the algorithm selects a policy $\boldsymbol{\pi}_t$ with the aim of maximizing the cumulative reward in that episode. We assume that the learner has no prior knowledge of the transition function, and can only learn about the MDP through interaction. The performance is measured in terms of the *regret*,

$$\mathfrak{R}_T = \sum_{t=1}^{K} (V_1^*(x_{1,t}) - V_1^{\boldsymbol{\pi}_k}(x_{1,t}))$$

where $T = KH$ is the total number of rounds and $x_{1,t} \sim \alpha$ is the initial state in episode $t$.

In tabular MDPs, the lower bound on the regret is $\Omega(H\sqrt{SAT})$ [24, 36, 25][3].Most optimistic algorithms are either model-optimistic or value-optimistic. Some notable model-optimistic approaches are UCRL2 [24] and REGAL [7] which have regret $\widetilde{O}(S\sqrt{H^3AT})$, and KL-UCRL [20, 48] and UCRL2-B [21], which have regret $\widetilde{O}(H\sqrt{S\Gamma AT})$ where $\Gamma \leq S$ is the maximal number of reachable states from any $(x,a) \in \mathcal{Z}$ and stage $h \in [H]$. These algorithms differ predominantly in the choice of distance and concentration bounds defining the set of feasible transition functions. Value-optimistic approaches often enjoy low regret at a cost of a more complex analysis. Some examples of these include UBEV [16] which has regret $\widetilde{O}(\sqrt{H^5SAT})$, and UCB-VI [6] which has regret $\widetilde{O}(H\sqrt{SAT})$ for large enough $T$ and $SA \geq H$, thus matching the lower bound. Other value optimistic algorithms achieving the optimal regret without requiring $SA \geq H$, are EULER [52] and ORLC [17]. These value optimistic approaches also often more clearly resemble empirically successful RL algorithms. We note that optimism has also been used in the model free setting (e.g. [25]), and that other non-optimistic approaches have also been successful at regret minimization (see e.g. [37, 3]). Other related works include [55, 41] which also use occupancy measures, [49] where optimistic linear programs are used, and [33, 48] which exploit duality in specific cases. Similar techniques have been developed in the context of robust learning in MDPs by [23] and [32].

For factored linear MDPs, all optimistic algorithms we are aware of are value-based, without a clear model-based interpretation: LSVI-UCB [26] uses dynamic programming and has regret $\widetilde{O}(\sqrt{d^3H^3T})$, while ELEANOR [54] has regret $\widetilde{O}(Hd\sqrt{T})$ but requires solving a complex optimization problem in each episode. The UC-MatrixRL algorithm [50] considers a different problem with two feature maps but is model-based with regret $\widetilde{O}(H^2d\sqrt{T})$. Non-optimistic approaches include [42, 53].

## 4   Optimism in Tabular Reinforcement Learning

We now present our main contribution: a general framework for designing, analyzing and implementing optimistic RL algorithms in episodic tabular MDPs. Our framework naturally extends the LPs in (1) and (2) to account for *uncertainty* about the transition function. We use confidence intervals for the transition functions to express uncertainty in the space of occupancy measures and maximize the expected reward over this set. Our key result shows that the dual of this optimization problem can be written in dynamic-programming form with added exploration bonuses, the size of which are determined by the shape of the primal confidence sets.

We define the uncertainty sets using confidence intervals around a reference transition function $\widehat{P}$. For a divergence measure $D(p, p')$ between probability distributions $p, p'$, define the confidence sets

$$\mathcal{P} = \left\{ \widetilde{P} \in \Delta : D\left(\widetilde{P}_h(\cdot|x,a), \widehat{P}_h(\cdot|x,a)\right) \leq \epsilon(x,a) \quad \forall (x,a) \in \mathcal{S} \times \mathcal{A}, h \in [H] \right\}, \quad (4)$$

where $\Delta$ is the set of valid transition functions. We assume that the divergence measure $D$ is jointly convex in its arguments so that $\mathcal{P}$ is convex, and that $D$ is positive homogeneous so for any $\alpha \geq 0$, $D(\alpha p, \alpha p') = \alpha D(p, p')$. Note that the distance $\|p - p'\|$ for any norm and all $f$-divergences satisfy these conditions [31]. Using $\mathcal{P}$, we modify (2) to get the optimistic primal optimization problem,

$$
\underset{\substack{q \in \mathcal{Q}(x_1) \\ \widetilde{P} \in \Delta}}{\text{maximize}} \quad \sum_{h=1}^{H} \langle q_{h,a}, r \rangle \quad \left|
\begin{array}{ll}
\text{subject to} \\
\sum_a q_{h+1,a} = \sum_a \widetilde{P}_{h,a}^{\mathsf{T}} q_{h,a} & \forall h \in [H] \\
D\left(\widetilde{P}_h(\cdot|x,a), \widehat{P}_h(\cdot|x,a)\right) \leq \epsilon(x,a) & \forall(x,a) \in \mathcal{Z},\, h \in [H]
\end{array}
\right. \tag{5}
$$

We pick $\epsilon$ such that $P \in \mathcal{P}$ with high probability. In this case, the above optimization problem returns an "optimistic" occupancy measure with higher expected reward than the true optimal policy. Unfortunately, the optimization problem in (5) is not convex due to the bilinear constraint $q_{h+1,a} = \sum_a \widetilde{P}_{h,a}^{\mathsf{T}} q_{h,a}$. Our main result below shows that it is still possible to obtain an equivalent value-optimistic formulation via Lagrangian duality and an appropriate reparametrization. We make use of the *conjugate* of the divergence $D$ defined for any function $z$, distribution $p'$ and threshold $\epsilon$ as

$$
D_* (z|\epsilon, p') = \max_{p \in \Delta} \left\{ \langle z, p - p' \rangle \,|\, D(p, p') \leq \epsilon \right\}.
$$

**Proposition 1.** *Let* $\mathrm{CB}_h(x,a) = D_*(V_{h+1}|\epsilon_h(x,a), \widehat{P}_h(\cdot|x,a))$ *and denote its vector representation by* $\mathrm{CB}_{h,a}$. *The optimization problem in* (5) *can be equivalently written as*

$$
\underset{V}{\text{minimize}}\ V_1(x_1) \quad \left|
\begin{array}{ll}
\text{subject to} \\
V_h \geq r_a + \widehat{P}_{h,a} V_{h+1} + \mathrm{CB}_{h,a} & \forall a \in \mathcal{A},\, h \in [H]
\end{array}
\right. \tag{6}
$$

*Proof sketch.* The full proof is in Appendix A.1. Here we outline the key ideas. To show strong duality, we reparameterize the problem as follows: define $J_h(x, a, x') = \widetilde{P}_h(x'|x,a)q_h(x,a)$ and note that due to homogeneity of $D$, the constraint on $\widetilde{P}$ is equivalent to $D(J_h(x,a,\cdot), \widehat{P}_h(\cdot|x,a)q_h(x,a)) \leq \epsilon_h(x,a)q_h(x,a)$, which is convex in $q$ and $J$. It is straightforward to verify the Slater condition for the resulting convex program, and thus strong duality holds for both parametrizations.

Letting $\mathcal{L}(q, \widetilde{P}; V)$ be the Lagrangian of (5) and using the non-negativity of $q$, the maximum of (5) is

$$
\min_{V} \max_{\substack{q \geq 0 \\ \widetilde{P} \in \mathcal{P}}} \mathcal{L}(q, \widetilde{P}; V) = \min_{V} \max_{q \geq 0} \left\{ \sum_{x,a,h} q_h(x,a) \left( \sum_y \widehat{P}_h(y|x,a) V_{h+1}(y) + r(x,a) - V_h(x) \right.\right.
$$

$$
\left.\left. + \max_{\widetilde{P}_h(\cdot|x,a) \in \mathcal{P}_h(x,a)} \sum_y \left( \widetilde{P}_h(y|x,a) - \widehat{P}_h(y|x,a) \right) V_{h+1}(y) \right) \right\}. \tag{7}
$$

Then, letting $\widehat{p} = \widehat{P}_h(\cdot|x,a)$, $\widetilde{p}(x') = \widetilde{P}_h(\cdot|x,a)$, and using the definition of $D$ and $D_*$, the inner maximum can be written as $\max_{\widetilde{p} \in \Delta} \{ \langle V_{h+1}, \widetilde{p} - \widehat{p} \rangle ; D(\widetilde{p}, \widehat{p}) \leq \epsilon_h(x,a) \} = D_*(V_{h+1}|\epsilon_h(x,a), \widehat{p})$. We then substitute this into (7) and use standard techniques to get the dual from the Lagrangian. $\square$

This result enables us to establish a number of important properties of the optimal solutions of the optimistic optimization problem (5). The following two propositions (proved in in Appendix A.2) highlight that optimal solutions to (5) are optimistic, bounded, and can be found by a dynamic-programming procedure. This implies that any model-optimistic algorithm that solves (5) in each episode is equivalent to value-optimistic algorithm using an appropriate choice of *exploration bonuses*.

**Proposition 2.** *Let* $V^+$ *be the optimal solution to* (6) *and* $\mathrm{CB}_h^+(x,a) = D_*(V_{h+1}^+|\epsilon(x,a), \widehat{P}_h)$. *Then, the optimal policy* $\pi^+$ *extracted from any optimal solution* $q^+$ *of the primal LP in* (5) *satisfies*

$$
V_h^+(x) = r(x, \pi_h^+(x)) + \mathrm{CB}_h^+(x, \pi_h^+(x)) + \sum_{y \in \mathcal{S}} \widehat{P}_h(y|x, \pi_h^+(x)) V_{h+1}^+(y) \quad \forall x \in \mathcal{S}, h \in [H]. \tag{8}
$$

**Proposition 3.** *If the true transition function* $P$ *satisfies the constraint in Equation* (5), *the optimal solution* $V^+$ *of the dual LP satisfies* $V_h^*(x) \leq V_h^+(x) \leq H - h + 1$ *for all* $x \in \mathcal{S}$.

## 4.1 Regret bounds for optimistic algorithms

We consider algorithms that, in each episode $t$, define the confidence sets $\mathcal{P}_t$ in (4) using some divergence measure $D$ and the reference model $\widehat{P}_{h,t}(x'|x,a) = \frac{N_{h,t}(x,a,x')}{N_{h,t}(x,a)}$ $\forall x, x' \in \mathcal{S}, a \in \mathcal{A}$. Here $N_{h,t}(x,a,x')$ is the total number of times that we have played action $a$ from state $x$ in stage $h$ and landed in state $x'$ up to the beginning of episode $t$, and $N_{h,t}(x,a) = \max\{\sum_{x'} N_{h,t}(x,a,x'), 1\}$. In episode $t$, the algorithm follows the optimistic policy $\boldsymbol{\pi}_t$ extracted from the solution of the primal optimistic problem in (5), or equivalently, the optimistic dynamic programming procedure in (6). The following theorem establishes a regret guarantee of the resulting algorithm:

**Theorem 4.** *On the event $\cap_{t=1}^K \{P \in \mathcal{P}_t\}$, the regret is bounded with probability at least $1-\delta$ as*

$$\mathfrak{R}_T \leq \sum_{t=1}^K \sum_{h=1}^H \left( \mathrm{CB}_{h,t}(x_{h,t}, \pi_{h,t}(x_{h,t})) + \mathrm{CB}_{h,t}^-(x_{h,t}, \pi_t(x_{h,t})) \right) + H\sqrt{2T\log(1/\delta)}$$

*where $\mathrm{CB}_{h,t}^-(x,a) = D_*(-V_{h+1,t}^+|\epsilon_{h,t}(x,a), \widehat{P}_{h,t})$ and $\mathrm{CB}_{h,t}(x,a) = D_*(V_{h+1,t}^+|\epsilon_{h,t}(x,a), \widehat{P}_{h,t})$.*

The proof is in Appendix A.3. While similar results are commonly used in the analysis of value-based algorithms [6, 16], the merit of Theorem 4 is that it is derived from a model-optimistic perspective, and thus cleanly separates the probabilistic and algebraic parts of the regret analysis. Indeed, proving the probabilistic statement that $P$ is in the confidence set is very simple in the primal space where our constraints are specified. Once this is established, the regret can be bounded in terms of the dual exploration bonuses. This simplicity of analysis is to be contrasted with the analyses of other value-optimistic methods that often interleave probabilistic and algebraic steps in a complex manner.

**Inflating the exploration bonus.** The downside of the optimistic dynamic-programming algorithm derived above is that the exploration bonuses may sometimes be difficult to calculate explicitly. Luckily, it is easy to show that the regret guarantees are preserved if we replace the bonuses by an easily-computed upper bound. This is helpful for instance when $D$ is defined as $D(p,p') = \|p - p'\|$, whence the conjugate can be simply bounded by the dual norm $\|V\|_*$. Formally, we can consider an *inflated conjugate* $D_*^\dagger$ satisfying $D_*^\dagger(f|\epsilon', \widehat{P}) \geq D_*(f|\epsilon, \widehat{P})$ for every function $f : \mathcal{S} \to [0, H]$, and obtain an optimistic value function by the following dynamic-programming procedure:

$$V_h^\dagger(x) = \max_a \left\{ \min \left\{ H - h + 1, r(x,a) + \widehat{P}_h(\cdot|x,a)V_{h+1}^\dagger + D_*^\dagger(V_{h+1}^\dagger|\epsilon'(x,a), \widehat{P}_h) \right\} \right\}, \quad (9)$$

with $V_{H+1}^\dagger(x) = 0 \, \forall x \in \mathcal{S}$. In this case, we need to clip the value functions since we can no longer use Proposition 3 to show they are bounded. The resulting value-estimates then satisfy $V_1^*(x_1) \leq V_1^+(x_1) \leq V_1^\dagger(x_1)$ with high probability, so we can bound the regret of this algorithm in the following theorem, whose proof is in Appendix A.4:

**Theorem 5.** *Let $D_*^\dagger(f|\epsilon', \widehat{P})$ be an upper bound on $D_*(f|\epsilon, \widehat{P})$ and $D_*(-f|\epsilon, \widehat{P})$ for every $f : \mathcal{S} \to [0, H]$, and, $\mathrm{CB}_{h,t}^\dagger(x,a) = D_*^\dagger(V_{h+1,t}^\dagger|\epsilon'_{h,t}(x,a), \widehat{P}_{h,t})$. Then, on the event $\cap_{t=1}^K \{P \in \mathcal{P}_t\}$, with probability greater than $1 - \delta$, the policy returned by the procedure in (9) incurs regret*

$$\mathfrak{R}_T \leq 2\sum_{t=1}^K \sum_{h=1}^H \mathrm{CB}_{h,t}^\dagger(x_{h,t}, \pi_{h,t}(x_{h,t})) + 4H\sqrt{2T\log(1/\delta)}.$$

**Examples.** Theorems 4 and 5 show that the key quantities governing the size of the regret are the conjugate distance and the confidence width $\epsilon$. This explicitly quantifies the impact of the choice of primal confidence set on the regret. We provide some example choices of divergences along with their conjugates, the best known confidence widths, and the resulting regret bounds in Table 4.1, with derivations in Appendix A.5. Many of these correspond to existing methods for which our framework suggests their first dynamic-programming implementation in the original state space $\mathcal{S}$, rather than the extended state-space which was traditionally used [24, 20, 33]. More generally, our framework captures any algorithm that defines confidence sets in terms of a norm or $f$-divergence, along with many others. It may also be possible to derive model-optimistic forms of value-optimistic methods. However, in this case care needs to be taken to show that the primal confidence sets are valid. For example, a variant of UCB-VI [6] can be derived from the divergence measure $\langle P - \widehat{P}, V_{h+1}^+ \rangle$, but the probabilistic analysis here is complicated due to the dependence between $\widehat{P}$ and $V_{h+1}^+$.

| Algorithm | Distance $D(p,\widehat{p})$ | $\epsilon$ | Conjugate $D_*^\dagger(V|\epsilon,\widehat{p})$ | Regret |
|---|---|---|---|---|
| UCRL2 [24]: | $\|p-\widehat{p}\|_1$ | $\sqrt{S/N}$ | $\epsilon\cdot\mathrm{span}\,(V)$ | $S\sqrt{H^3AT}$ |
| UCRL2B [21]: | $\max_x\frac{(p(x)-\widehat{p}(x))^2}{\widehat{p}(x)}$ | $1/N$ | $\sum_x\sqrt{\epsilon\widehat{p}(x)}|V(x)-\widehat{p}V|$ | $H\sqrt{S\Gamma AT}$ |
| KL-UCRL[4]: | $\mathrm{KL}(p,\widehat{p})$ | $S/N$ | $\sqrt{(\epsilon+(1-\sum_y\widehat{p}(y)))\widehat{\mathbb{V}}(V)}$ | $HS\sqrt{AT}$ |
| $\chi^2$-UCRL[5] | $\sum_x\frac{(p(y)-\widehat{p}(y))^2}{\widehat{p}(y)}$ | $S/N$ | $\sqrt{\epsilon\widehat{\mathbb{V}}(V)}$ | $HS\sqrt{AT}$ |

Table 1: Various algorithms in our framework. For all algorithms except UCRL2, we use $\widehat{P}_{h,t}^+(y|x,a)=\frac{\max 1,N_{h,t}(x,a,y)}{N_{h,t}(x,a)}$ as the base measure to avoid division by 0, for UCRL2, we use $\widehat{P}(y|x,a)$. We denote $\widehat{\mathbb{V}}(V)=\sum_x\widehat{p}(x)\,(V(x)-\langle\widehat{p},V\rangle)^2$. The third column gives scaling of the confidence width in terms of $S$ and the number of sample transitions $N$. The fourth column gives a tractable upper bound on the value of the conjugate. The last column gives the the regret bound derived from Theorem 5 (up to logarithmic factors) with exploration bonus defined from the inflated conjugate and the smallest value of $\epsilon$ that guarantees $\cap_{t=1}^K\{P\in\mathcal{P}_t\}$ w.h.p.

# 5 Optimism with realizable linear function approximation

We now extend our framework to factored linear MDPs, where all currently known algorithms are value-optimistic. We provide the first model-optimistic formulation by modeling uncertainty about the MDP in the primal LP involving occupancy measures in (3). All proofs are in Appendix B.

A key challenge in this setting is that the uncertainty can no longer be expressed using distance metrics in the state space, since this could lead to trivially large confidence sets[6]. Instead, we define confidence sets in terms of a distance that takes the linear structure into account. These are centered around a reference model $\widehat{P}$ defined for each $h,a$ as $\widehat{P}_{h,a}=\Phi\widehat{M}_{h,a}$ for some $d\times S$ matrix $\widehat{M}_{h,a}$. We consider reference models implicitly defined by the LSTD algorithm [12, 28, 38]. In episode $t$, let $\Sigma_{h,a,t}=\sum_{k=1}^t\mathbb{I}_{\{a_{h,k}=a\}}\varphi(x_{h,k})\varphi^\mathsf{T}(x_{h,k})+\lambda I$ for some $\lambda\geq 0$, and $\mathbf{e}_x$ be the unit vector in $\mathbb{R}^S$ corresponding to state $x$. Then, our reference model in episode $t$ is defined for each action $a$ as

$$\widehat{M}_{h,a,t}=\Sigma_{h,a,t-1}^{-1}\sum_{k=1}^{t-1}\mathbb{I}_{\{a_{h,k}=a\}}\varphi(x_{h,k})\mathbf{e}_{x_{h+1,k}}. \tag{10}$$

Finally, the weight matrix in the LP formulation (3) is chosen as $W_{h,a,t}=\sum_{k=1}^t\mathbb{I}_{\{a_{h,k}=a\}}\mathbf{e}_{x_{h,k}}\mathbf{e}_{x_{h,k}}^\mathsf{T}$, so that $\Phi^\mathsf{T}W_{h,a,t}\Phi=\Sigma_{h,a,t}-\lambda I$. We establish the following important technical result:

**Proposition 6.** *Consider the reference model $\widehat{P}_{h,a,t}=\Phi\widehat{M}_{h,a,t}$ with $\widehat{M}_{h,a,t}$ defined in Equation* (10). *Then, for any fixed function $g:\mathcal{S}\to[-H,H]$, the following holds with probability at least $1-\delta$:*

$$\left\|\left(M_{h,a,t}-\widehat{M}_{h,a,t}\right)g\right\|_{\Sigma_{h,a,t-1}}\leq H\sqrt{d\log\left(\frac{1+tR^2/\lambda}{\delta}\right)}+C_PH\sqrt{\lambda d}.$$

The proof is based on the fact that for a fixed $g$, $\left(M_{h,a,t}-\widehat{M}_{h,a,t}\right)g$ is essentially a vector-valued martingale. Our main contribution in this setting is to use this result to identify two distinct ways of deriving tight confidence sets that incorporate optimism into (3). Both approaches use the optimistic parametric Bellman (OPB) equations with some exploration bonus $\mathrm{CB}_{h,t}(x,a)$ (defined later):

$$\theta_{h,a,t}^+=\rho_a+\Sigma_{h,a,t-1}^{-1}\sum_{k=1}^{t-1}\mathbb{I}_{\{a_{h,k}=a\}}\varphi(x_{h,k})V_{h+1,t}^+\left(x_{h+1,k}\right)$$
$$V_{h,t}^+(x)=\max_a\left\{\left(\Phi\theta_{h,a,t}^+\right)(x)+\mathrm{CB}_{h,t}(x,a)\right\} \tag{11}$$

Both bonuses we derive can be upper-bounded by $\mathrm{CB}^{\dagger}_{h,t}(x,a) = C(d)\,\|\varphi(x)\|_{\Sigma^{-1}_{h,a,t-1}}$ for some $C(d) > 0$. Then, one can apply a variant Theorem 5 to bound the regret of both algorithms in terms of the sum of these inflated exploration bonuses, amounting to a total regret of $\widetilde{O}(C(d)\sqrt{dHT})$.

## 5.1 Optimism in state space through local confidence sets

Our first approach models the uncertainty locally in each state-action pair $x, a$ using some distance metric $D$ between transition functions. We consider the following optimization problem:

$$
\begin{array}{ll}
\underset{q \in \mathcal{Q}(x_1),\omega}{\text{maximize}} & \text{subject to} \\
\sum_{h=1}^{H} \sum_a \langle W_{h,a}\Phi\omega_{a,h}, r_a \rangle & \begin{array}{ll}
\sum_a q_{h+1,a} = \sum_a \widetilde{P}_{h,a}W_{h,a}\Phi\omega_{h,a} & \forall h \in [H] \\
\Phi^\intercal q_{h,1} = \Phi^\intercal W_{h,a}\Phi\omega_{h,a} & \forall a \in \mathcal{A}, h \in [H] \\
D\left(\widetilde{P}_h(\cdot|x,a), \widehat{P}_h(\cdot|x,a)\right) \leq \epsilon_h(x,a) & \forall (x,a) \in \mathcal{Z}, h \in [H]
\end{array}
\end{array} \tag{12}
$$

As in the tabular case, (12) can be reparametrized so that the constraint set is convex, allowing us to appeal to Lagrangian duality to get an equivalent formulation as shown in the following proposition.

**Proposition 7.** *The optimization problem* (12) *is equivalent to solving the optimistic Bellman equations* (11) *with the exploration bonus defined as* $\mathrm{CB}_h(x,a) = D^*(V^+_{h+1}|\epsilon_h(x,a), \widehat{P}_h(\cdot|x,a))$.

Taking the form of $V^+_h$ into account, in episode $t$, we define our confidence sets as in (4) with

$$
D\left(\widetilde{P}_{h,t}(\cdot|x,a), \widehat{P}_{h,t}(\cdot|x,a)\right) = \sup_{g \in \mathcal{V}_{h+1,t}} \sum_{x'} \left(\widetilde{P}_{h,t}(x'|x,a) - \widehat{P}_{h,t}(x'|x,a)\right) g(x') \tag{13}
$$

and $\widehat{P}_{h,a,t} = \Phi\widehat{M}_{h,a,t}$ where $\mathcal{V}_{h+1,t}$ is the set of value functions that can be produced by solving the OPB equations (11). For any choice of $\epsilon_t$, $\mathrm{CB}_{h,t}(x,a) \leq \epsilon_{h,t}(x,a)$, so one can simply use the bonus $\mathrm{CB}^{\dagger}_{h,t}(x,a) = \epsilon_{h,t}(x,a)$. The following theorem bounds the regret for an appropriate choice of $\epsilon_t$

**Theorem 8.** *The choice* $\epsilon_{h,t}(x,a) = C\,\|\varphi(x)\|_{\Sigma^{-1}_{h,a,t-1}}$ *with* $C = \widetilde{O}(Hd)$ *guarantees that the transition model $P$ is feasible for* (12) *in every episode $t$ with probability* $1-\delta$. *The resulting optimistic algorithm with exploration bonus* $\mathrm{CB}^{\dagger}_{h,t}(x,a) = \epsilon_{h,t}(x,a)$ *has regret bounded by* $\widetilde{O}(\sqrt{H^3 d^3 T})$.

This algorithm coincides with the LSVI-UCB method of [26] and our performance guarantee matches theirs. The advantage of our result is a simpler analysis allowed by our model-optimistic perspective.

## 5.2 Optimism in feature space through global constraints

Our second approach exploits the structure of the reference model (10), and constrains $\widetilde{P}_a$ through global conditions on $\widetilde{M}_a$. We define $\mathcal{P}_t$ using the distance metric suggested by Proposition 6 as

$$
D(\widetilde{M}_{h,a}, \widehat{M}_{h,a}) = \sup_{f \in \mathcal{V}_{h+1}} \left\|(\widetilde{M}_{h,a} - \widehat{M}_{h,a})f\right\|_{\Sigma_{h,a}} \leq \epsilon_{h,a} \tag{14}
$$

for $\mathcal{V}_{h+1}$ as in (13) and some $\epsilon_{h,a} > 0$. We then consider the following optimization problem:

$$
\begin{array}{ll}
\underset{\substack{q \in \mathcal{Q}(x_1), \\ \omega, \widetilde{M}}}{\text{maximize}} \quad \sum_{h=1}^{H} \sum_a \langle W_{h,a}\Phi\omega_{h,a}, r_a \rangle & \begin{array}{ll}
\text{subject to} & \\
\sum_a q_{h+1,a} = \sum_a \widetilde{P}^\intercal_{h,a}W_{h,a}\Phi\omega_{h,a} & \forall h \in [H] \\
\Phi^\intercal q_{h,a} = \Phi^\intercal W_{h,a}\Phi\omega_{h,a} & \forall a \in \mathcal{A}, h \in [H] \\
D(\widetilde{M}_{h,a}, \widehat{M}_{h,a}) \leq \epsilon_{h,a} & \forall a \in \mathcal{A}, h \in [H].
\end{array}
\end{array} \tag{15}
$$

Unfortunately, directly constraining $\widetilde{M}$ leads to an optimization problem that, unlike in the other settings, cannot easily be re-written as an convex problem exhibiting strong duality. Nevertheless, for a fixed $\widetilde{M}$, the value of (15) is equivalent to $G(\widetilde{M}) = V^+_1(x_1)$ where $V^+$ solves the OPB equations (11) with $\mathrm{CB}_h(x,a) = \langle\varphi(x), (\widetilde{M}_{h,a} - \widehat{M}_{h,a})V_{h+1}\rangle$. Let $\mathcal{M} = \{\widetilde{M} \in \mathbb{R}^{d \times S} : D(\widetilde{M}, \widehat{M}) \leq \epsilon\}$, then, we can re-write (15) as maximizing $G(\widetilde{M})$ over $\widetilde{M} \in \mathcal{M}$. Exploiting this we provide a more tractable version of the optimization problem, and bound the regret of the resulting algorithm, below:

**Theorem 9.** *Define the function* $G'(B) = V^+_1(x_1)$ *with* $V^+$ *the solution of the OPB equations* (11) *with exploration bonus* $\mathrm{CB}_h(x,a) = \langle\varphi(x), B_{h,a}\rangle$ *and let* $\mathcal{B}_t = \left\{B : \|B_{h,a}\|_{\Sigma_{h,a,t-1}} \leq \epsilon_{h,a,t}\right\}$ *for all episodes* $t \in [K]$. *Then,* $\max_{B \in \mathcal{B}_t} G'(B) \geq \max_{\widetilde{M} \in \mathcal{M}_t} G(\widetilde{M})$ *and the optimistic algorithm with exploration bonuses corresponding to the optimal solutions* $B^{\dagger}_t$ *has regret bounded by* $\widetilde{O}(d\sqrt{H^3 T})$.

The algorithm suggested in this theorem essentially coincides with the ELEANOR method proposed recently in [54], and our guarantees match theirs under our realizability assumption. Our model-based perspective suggests that the problem of implementing ELEANOR is inherently hard: the form of the primal optimization problem reveals that $G'(B)$ is a convex function of $B$, and thus its maximization over a convex set is intractable in general. Note that the celebrated LinUCB algorithm for linear bandits must solve the a similar convex maximization problem [15, 1]. As in linear bandits, it remains an open question to get regret $\widetilde{O}(Hd\sqrt{T})$ with a computationally efficient algorithm.

## 6 Conclusion

We have provided a new framework unifying model-optimistic and value-optimistic approaches for episodic reinforcement learning. These results demonstrate that many desirable features are enjoyed by both approaches.

In the tabular setting, we provided improved implementations and analyses of a general class of model-optimistic algorithms. While these results demonstrate the strength and flexibility of the model-based perspective, our regret bounds feature an additional factor of $\sqrt{S}$ on top of the minimax optimal bounds, which has been eliminated by value-optimistic methods [6, 16]. We believe that in order to recover these algorithms in our framework with improved regret bounds in the tabular setting, it may be necessary to consider more sophisticated (potentially non-local) confidence sets on the transition functions. In this paper, we have shown that this is indeed the case for factored linear MDPs. Here using global constraints in a model-optimistic algorithm was shown to be equivalent to a state of the art value-optimistic algorithm. This leads us to believe that a similar approach may enable us to derive model-optimistic algorithms that are equivalent to state of the art value-optimistic algorithms in the tabular setting, although we leave this as future work.

Finally, we note that our structural results concerning equivalence of model constraints and reward perturbations may have impact beyond the particular problem we consider in this paper. One immediate generalization of our results beyond regret minimization is to the PAC MDP setting, where the same techniques can be applied to draw parallels between model-based and model-free algorithms [16]. Beyond optimistic exploration, uncertainty sets and reward perturbations are also broadly used in the context of robust optimization in MDPs, and in fact Iyengar [23] has previously established connections between these concepts in a way similar to our work. It remains to be seen if these connections can be further explored and if establishing a convex-optimization formulation of robust MDPs is possible in a way similar to the proof of our Proposition 1. Lastly, it is straightforward to extend our framework for infinite-horizon MDPs, although analyzing the regret of the resulting algorithms remains challenging. We leave pursuing these exciting directions for future research.

## Broader Impact

The results presented in this paper are largely theoretical. We define a class of algorithms which are theoretically well understood, but also benefit from a computationally efficient implementation. The framework provided in this paper is very general so, in principle, any algorithm which fits into the framework could be applied to any reinforcement learning problem in a tabular or factored linear MDP. Consequently, as for any reinforcement learning algorithm, there is the potential for algorithms developed using the ideas presented in this paper to be applied in settings which have negative societal impacts, or in settings where the reward function is not well specified leading to undesirable behaviors.

## Acknowledgments

G. Neu was supported by "la Caixa" Banking Foundation through the Junior Leader Postdoctoral Fellowship Programme, a Google Faculty Research Award, and the Bosch AI Young Researcher Award.

## Footnotes

[2]The extension to unknown rewards is fairly straightforward using upper confidence bounds on $r$

[3]The extra $\sqrt{H}$ due to having a different $P_h$ per stage. We use $\widetilde{O}(\cdot)$ to denote order up to logarithmic terms.

[4]We consider the un-normalized KL-divergence, $\mathrm{KL}(p,\widehat{p})=\sum_x p(x)\log\frac{p(x)}{\widehat{p}(x)}+\sum_x(\widehat{p}(x)-p(x))$. The original KL-UCRL algorithm, [20, 48] considers the reverse (normalized) KL-divergence. This also fits into our framework. See Appendix A.5.3 for details.

[5][33] also use a $\chi^2$-divergence but require $\widetilde{P}(x)>p_0$ for some $p_0$ if $\widetilde{P}(x)>0$ making $\mathcal{P}$ non-convex.

[6]E.g., for the total variation distance, concentration bounds scale with $\sqrt{S}$ which is potentially unbounded.

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
