[Supplementary Material]

# Appendix

## A   Proofs of Results for Tabular Setting

We prove here the results of Section 4. For ease of exposition, we restate the results before proving them. For convenience, we introduce the confidence set for every state $x \in \mathcal{S}$, action $a \in \mathcal{A}$ and stage $h \in [H]$,

$$\mathcal{P}_h(x,a) = \left\{ \widetilde{P}_h(\cdot|x,a) \in \Delta : D\left( \widetilde{P}_h(\cdot|x,a), \widehat{P}_h(\cdot|x,a) \right) \leq \epsilon(x,a) \right\} \tag{16}$$

and note that $\widetilde{P} \in \mathcal{P}$ if $\widetilde{P}_h(\cdot|x,a) \in \mathcal{P}_h(x,a)$ for all $x, a, h$

The following lemma will be useful in several of the proofs.

**Lemma 10.** *The primal and dual optimization problems in* (5) *and* (6) *exhibit strong duality. Consequently the Karush-Kuhn-Tucker (KKT) conditions hold, and in particular, complementary slackness holds.*

*Proof.* We first show that the optimization problem in (5) exhibits strong duality. For this, it is helpful to consider a reparameterization where we introduce the variables $J_h(x,a,x') = \widetilde{P}_h(x'|x,a)q_h(x,a)$, so that the constraint $D\left(\widetilde{P}_h(\cdot|x,a), \widehat{P}_h(\cdot|x,a)\right) \leq \epsilon(x,a)$ can be rewritten as $D\left(J_h(x,a,\cdot), \widehat{P}_h(\cdot|x,a)q_h(x,a)\right) \leq \epsilon_h(x,a)q_h(x,a)$, which is convex in $J$ and $q$. The two constraints are clearly equivalent due to positive homogeneity of $D$ for $q_h(x,a) > 0$. For $q_h(x,a) = 0$, recall that in the original formulation, $\widetilde{P}_h(\cdot|x,a)$ only appears multiplied by $q_h(x,a)$, so when $q_h(x,a) = 0$, the choice of $\widetilde{P}_h(\cdot|x,a)$ is arbitrary and we can replace the constraint $D\left(\widetilde{P}_h(\cdot|x,a), \widehat{P}_h(\cdot|x,a)\right) \leq \epsilon(x,a)$ with $D\left(J_h(x,a,\cdot), \widehat{P}_h(\cdot|x,a)q_h(x,a)\right) \leq \epsilon_h(x,a)q_h(x,a)$ without affecting the optimal solution. This implies that the optimization problem in (5) can be equivalently written as

$$\underset{q \in \mathcal{Q}(x_1), J}{\text{maximize}} \sum_{x,a,h} q_h(x,a)r(x,a) \tag{17}$$

$$\text{Subject to } \sum_a q_h(x,a) = \sum_{x',a'} J_{h-1}(x',a',x) \qquad \forall x \in \mathcal{S}, h \in [H]$$

$$D\left( J_h(x,a,\cdot), \widehat{P}_h(\cdot|x,a)q_h(x,a) \right) \leq \epsilon_h(x,a)q_h(x,a) \qquad \forall (x,a) \in \mathcal{Z}, h \in [H]$$

$$\sum_{x'} J_h(x,a,x') = q_h(x,a) \qquad \forall (x,a) \in \mathcal{Z}, h \in [H]$$

$$J_h(x,a,x') \geq 0 \qquad \forall x, x' \in \mathcal{S}, a \in \mathcal{A}, h \in [H].$$

In this formulation, there is only one non-linear constraint, and by our assumption that $D$ is convex in both of its arguments, this constraint is convex in $J$ and $q$. Moreover, $\widehat{J}_h(x,a,x') = \widehat{P}_h(x'|x,a)q_h(x,a)$ satisfies this constraint for any $q_h(x,a)$, and in particular, if $q_h(x,a)$ is the occupancy measure induced by any policy $\pi$ in the MDP with transition function $\widehat{P}$, then $q_h(x,a)$ and $J_h(x,a,a')$ are feasible solutions to the primal. Hence, the Slater conditions are satisfied, and thus the optimization problem exhibits strong duality (see e.g. [11]). We can then write the dual of the optimization problem in (17) as

$$\max_{(q,M) \in \mathcal{C}_1} \min_{V,\gamma} \left\{ \sum_{x,a,h} q_h(x,a)(V_h(x) - \gamma_h(x,a) + r(x,a) + V_1(x_1) \tag{18} \right.$$

$$\left. + \sum_{x,a,x',h} J_h(x,a,x')(V_{h+1}(x') + \gamma_h(x,a)) \right\},$$

where $\mathcal{C}_1 = \left\{ q, J : D(J_h(x,a,\cdot), \widehat{P}_h(\cdot|x,a)q_h(x,a)) \leq \epsilon_h(x,a)q_h(x,a) \ (\forall x,a) \right\}$. Then, we can use the reverse reparameterization to rewrite this in terms of $\widetilde{P}_h(x'|x,a) = J_h(x,a,x')/q_h(x,a)$,

noting that $\widetilde{P}_h(\cdot|x,a)$ is a valid probability density by constraints on $J, q$. We get,

$$\max_{q,\widetilde{P}\in\mathcal{P}}\min_{V,\gamma}\left\{\sum_{x,a,h}q_h(x,a)(-V_h(x)-\gamma_h(x,a)+r(x,a))+V_1(x_1)\right.$$

$$\left.+\sum_{x,a,x',h}\widetilde{P}_h(x'|x,a)q_h(x,a)(V_{h+1}(x')+\gamma_h(x,a))\right\}$$

$$=\max_{q,\widetilde{P}\in\mathcal{P}}\min_{V}\left\{\sum_{x,a,h}q_h(x,a)\left(-V_h(x)+r(x,a)+\sum_{x'}\widetilde{P}_h(x'|x,a)V_{h+1}(x')\right)+V_1(x_1)\right\},$$

(19)

where $\mathcal{P}=\left\{\widetilde{P}\in\Delta:D(\widetilde{P}_h(\cdot|x,a),\widehat{P}_h(\cdot|x,a))\le\epsilon_h(x,a)\ \ (\forall x,a,h)\right\}$, and the last equality follows since $\sum_y\widetilde{P}_h(y|x,a)=1$. This is the Lagrangian dual form of the original optimization problem we considered. Let $\mathrm{OBJ}(a)$ denote the objective function of the optimization problem in equation $(a)$. It then follows that,

$$\mathrm{OBJ}(5)=\mathrm{OBJ}(17)=\mathrm{OBJ}(18)=\mathrm{OBJ}(19)$$

and so strong duality holds for the problem in (5). Thus, by standard results (e.g., [11, Section 5.5.3]), we conclude that the KKT conditions are satisfied by $(q^+,\widetilde{P}^+,V^+)$, the optimal solutions to the primal and dual. As a consequence, complementary slackness also holds. This concludes the proof. □

## A.1 Duality Result

**Proposition 1.** *Let* $\mathrm{CB}_h(x,a)=D_*(V_{h+1}|\epsilon_h(x,a),\widehat{P}_h(\cdot|x,a))$ *and denote its vector representation by* $\mathrm{CB}_{h,a}$. *The optimization problem in* (5) *can be equivalently written as*

$$\underset{V}{minimize}\ V_1(x_1)\ \bigg|\ \begin{array}{l}subject\ to\\ V_h\ge r_a+\widehat{P}_{h,a}V_{h+1}+\mathrm{CB}_{h,a}\quad\forall a\in\mathcal{A},h\in[H]\end{array}\qquad(6)$$

*Proof.* It will be helpful to write the primal optimization problem as

$$\underset{q\in\mathcal{Q}(x_1),\widetilde{P},\kappa}{\text{maximize}}\ \sum_{x,a,h}q_h(x,a)r(x,a)$$

Subject to

$$\sum_a q_h(x,a)=\sum_{x',a'}\widehat{P}_h(x|x',a')q_h(x',a')+\sum_{x',a'}\kappa_h(x',a',x)q_h(x',a')\qquad\forall x\in\mathcal{S},h\in[H]$$

$$\kappa_h(x,a,x')=\widetilde{P}_h(x'|x,a)-\widehat{P}_h(x'|x,a)\qquad\qquad\forall x,x'\in\mathcal{S},a\in\mathcal{A},h\in[H]$$

$$D\left(\widehat{P}_h(\cdot|x,a),\widehat{P}_h(\cdot|x,a)\right)\le\epsilon_h(x,a)\qquad\qquad\forall(x,a)\in\mathcal{Z},h\in[H]$$

$$\sum_{x'}\kappa_h(x,a,x')=0\qquad\qquad\forall(x,a)\in\mathcal{Z},h\in[H].$$

By Lemma 10, we know that this problem exhibits strong duality. We then consider the partial Lagrangian of the above problem without the constraints on $\widetilde{P}$, which yields

$$\mathcal{L}(q,\kappa;V)=\sum_{x,a,h}q_h(x,a)\left(\sum_y\widehat{P}_h(y|x,a)V_{h+1}(y)+\sum_y\kappa_h(x,a,y)V_{h+1}(y)+r(x,a)-V_h(x)\right)+V_1(x_1)$$

For $\mathcal{P}$ defined in (4), we know that the optimal value of the objective function of the primal optimization problem is given by the Lagrangian relaxation,

$$\min_V\ \max_{q\ge0,\kappa,\widetilde{P}\in\mathcal{P}}\ \mathcal{L}(q,\kappa;V).$$

To proceed, we fix a $V$ and consider the inner maximization problem. By definition of $\kappa_h(x, a, x') = \widetilde{P}_h(x'|x, a) - \widehat{P}_h(x'|x, a))$, we can write

$$\max_{q \geq 0, \kappa, \widetilde{P} \in \mathcal{P}} \mathcal{L}(q, \kappa; V)$$

$$= \max_{q \geq 0, \kappa, \widetilde{P} \in \mathcal{P}} \sum_{x,a,h} q_h(x, a) \left( \sum_y \widehat{P}_h(y|x, a) V_{h+1}(y) + \sum_y \kappa_h(x, a, y) V_{h+1}(y) + r(x, a) - V_h(x) \right)$$

$$+ V_1(x_1)$$

$$= \max_{q \geq 0} \sum_{x,a,h} q_h(x, a) \left( \sum_y \widehat{P}_h(y|x, a) V_{h+1}(y) + \max_{\substack{\kappa_h(x,a,\cdot) \\ \widetilde{P}_h(\cdot|x,a) \in \mathcal{P}_h(x,a)}} \sum_y \kappa(x, a, y) V_{h+1}(y) + r(x, a) \right.$$

$$\left. - V_h(x) \right) + V_1(x_1)$$

$$= \max_{q \geq 0} \sum_{x,a,h} q_h(x, a) \left( \sum_y \widehat{P}_h(y|x, a) V_{h+1}(y) + D_*(V_{h+1}|\epsilon_h(x, a), \widehat{P}_h(\cdot|x, a)) + r(x, a) - V_h(x) \right)$$

$$+ V_1(x_1), \tag{20}$$

where $\mathcal{P}_h(x, a)$ is the set in (16). The second equality crucially uses that $q_h(x, a) \geq 0$ and the last equality follows from the definition of the conjugate $D_*$:

$$\max_{\kappa_h(x,a,\cdot), \widetilde{P}_h(\cdot|x,a)) \in \mathcal{P}_h(x,a)} \sum_y \kappa_h(x, a, y) V_{h+1}(y)$$

$$= \max_{\widetilde{P}_h(\cdot|x,a) \in \Delta} \left\{ \langle \widetilde{P}_h(\cdot|x, a) - \widehat{P}_h(\cdot|x, a), V_{h+1} \rangle; D(\widetilde{P}_h(\cdot|x, a), \widehat{P}_h(\cdot|x, a)) \leq \epsilon_h(x, a) \right\}$$

$$= D_*(V_{h+1}|\epsilon_h(x, a), \widehat{P}_h(\cdot|x, a)).$$

We then optimize the expression in (20) with respect to $q$ and $V$ using an adaptation of techniques used for establishing LP duality between the original problems (1) and (2). Specifically, let $g(V) = \max_q \mathcal{L}(q; V)$ and note that by (20), the Lagrangian no longer depends on $\kappa$ or $\widetilde{P}$. Then, define $\eta_h(x, a) = \sum_y \widehat{P}_h(y|x, a) V_{h+1}(y) + D_*(V_{h+1}|\epsilon_h(x, a), \widehat{P}_h(\cdot|x, a)) + r(x, a) - V_h(x)$ for all $x, a, h$ and observe that

$$g(V) = V_1(x_1) + \max_q \sum_{x,a,h} q_h(x, a) \eta_h(x, a) = \begin{cases} V_1(x_1) & \text{if } \eta_h(x, a) \leq 0 \quad \forall x, a, h \\ \infty & \text{otherwise.} \end{cases}$$

Thus, we can then write the dual optimization problem of minimizing $g(V)$ with respect to $V$ as

$$\underset{V}{\text{minimize}} \quad V_1(x_1)$$

$$\text{Subject to} \quad V_h(x) \geq r(x, a) + \sum_y \widehat{P}_h(y|x, a) V_{h+1}(y) + D_*(V_{h+1}|\epsilon_h(x, a), \widehat{P}_h(\cdot|x, a)).$$

This proves the proposition. $\qquad\square$

## A.2  Properties of the Optimal Solutions

In this section we prove Propositions 2 and 3. In order to prove Proposition 2, we first need the following result which gives the form of the optimal solution to the dual in Equation (6).

**Lemma 11.** *The solution to the dual in (6) is given by*

$$V_h^+(x) = \max_{a \in \mathcal{A}} \left\{ r(x, a) + \text{CB}_h(x, a) + \sum_{y \in \mathcal{S}} \widehat{P}_h(y|x, a) V_{h+1}^*(y) \right\} \tag{21}$$

*where we use the notation* $\text{CB}_h(x, a) = D_*(V_{h+1}|\epsilon_h(x, a), \widehat{P}_h(\cdot|x, a)))$.

*Proof.* The structure of the constraints on $V_h(x)$ in (6) and the definition of $\mathrm{CB}_h(x,a)$ mean that $V_h^+(x)$ can be determined using only the values of $V_l^+$ for $l \geq h+1$. Hence, we can prove the result by backwards induction on $h = H, \ldots, 1$. For the base case, when $h = H$, the constraint in the dual is

$$V_H(x) \geq r(x,a) + \mathrm{CB}_H(x,a) \quad \forall x \in \mathcal{S}, a \in \mathcal{A}.$$

In order to minimize $V_H(x)$, we set $V_H^+(x) = \max_{a \in \mathcal{A}}\{r(x,a) + \mathrm{CB}_H(x,a)\}$ for all $x \in \mathcal{S}$. Now assume that for stage $h+1$, the optimal value of $V_{h+1}^+(x)$ is given by (21). Then, when considering stage $h$, we wish to set $V_h^+(x)$ as small as possible. By the inductive hypothesis, we know it is optimal to set $V_{h+1}(x) = V_{h+1}^+(x)$, and we know that $\mathrm{CB}_h(x,a)$ has been defined using only terms from stage $h+1$ and is minimal. Consequently, the RHS of the constraint in (6) is minimized for any $(x,a,h)$ by setting $V_{h+1} = V_{h+1}^+$. This means that the minimal value of $V_h$ is given by (21). Hence the result holds for all $h = 1, \ldots, H$, and so considering $h = 1$ and initial state $x_1$, we can conclude that $V^+$ is the optimal solution to the LP in (6). $\qquad\square$

We now prove Proposition 2.

**Proposition 2.** *Let $V^+$ be the optimal solution to* (6) *and* $\mathrm{CB}_h^+(x,a) = D_*(V_{h+1}^+|\epsilon(x,a), \widehat{P}_h)$. *Then, the optimal policy $\pi^+$ extracted from any optimal solution $q^+$ of the primal LP in* (5) *satisfies*

$$V_h^+(x) = r(x, \pi_h^+(x)) + \mathrm{CB}_h^+(x, \pi_h^+(x)) + \sum_{y \in \mathcal{S}} \widehat{P}_h(y|x, \pi_h^+(x))V_{h+1}^+(y) \quad \forall x \in \mathcal{S}, h \in [H]. \quad (8)$$

*Proof.* By Lemma 11, we know that the optimal solution to the dual in (6) is given by

$$V_h^+(x) = \max_{a \in \mathcal{A}} \left\{ r(x,a) + \mathrm{CB}_h(x,a) + \sum_{y \in \mathcal{S}} \widehat{P}_h(y|x,a)V_{h+1}^+(y) \right\}. \quad (22)$$

We then proceed by considering the case where the right hand side of the expression in (22) has a unique maximizer. In this case, let

$$a_h^*(x) = \operatorname*{arg\,max}_{a \in \mathcal{A}} \left\{ r(x,a) + \mathrm{CB}_h(x,a) + \sum_{y \in \mathcal{S}} \widehat{P}_h(y|x,a)V_{h+1}^+(y) \right\}.$$

Since $a_h^*(x)$ is the unique maximizer of this expression, it follows that, for a fixed $x, h$, the constraint in (6) is only binding for one $a \in \mathcal{A}$, namely $a_h^*(x)$. By Lemma 10, we know that complementary slackness holds for this problem. Then, using complementary slackness, it follows that only one of the primal variables is non-zero. In particular, for a fixed state $x$ and stage $h$, $q_h^+(x,a) = 0$ for all $a \neq a_h^*(x), x' \in \mathcal{S}$. Consequently, $\pi^+(x) = a_h^*(x)$ and so the policy induced by $q^+, \pi^+$, will only have non-zero probability of playing the action which maximize the right hand side of (22).

We now consider the case where there are multiple maximizers of the right hand side of (22). Let $a_h^1(x), \ldots, a_h^m(x)$ denote the $m$ maximizers. By a similar argument to the previous case, we know that for a fixed $x \in \mathcal{S}$ and $h \in [H]$, the constraint in (6) is only binding for $a = a_h^i(x)$ for some $i \in [m]$. Then, by complementary slackness, it follows that $q_h^+(x,a) = 0$ for all $a \neq a_i^*(x)$ for $i \in [m]$, and so the only non-zero values of $q_h^+(x,a)$ can occur for $a = a_h^i(x)$ for some $i \in [m]$. The action chosen from state $x$ by policy $\pi^+$ must be one of the actions for which $q_h^+(x,a) > 0$ by properties of the relationship between occupancy measures and policies. Hence, $\pi^+(x) = a_h^i(x)$ for some $i \in [m]$, and so equation (8) must hold. $\qquad\square$

**Proposition 3.** *If the true transition function $P$ satisfies the constraint in Equation* (5)*, the optimal solution $V^+$ of the dual LP satisfies $V_h^*(x) \leq V_h^+(x) \leq H - h + 1$ for all $x \in \mathcal{S}$.*

*Proof.* We begin by proving that if $P \in \mathcal{P}$, then $V_h^*(x) \leq V_h^+(x)$.

Let $q^*$ be the occupancy measure corresponding to the optimal policy $\pi^*$ under $P$. Then, if $P \in \mathcal{P}$, then $P$ must feasible for the primal in (5), and so it must be the case that

$$\sum_{x,a} \sum_{h=1}^{H} r(x,a)q_h^*(x,a) \leq \sum_{x,a} \sum_{h=1}^{H} r(x,a)q_h^+(x,a),$$

where $q^+$ is the optimal solution to the LP in (5). Considering the LHS of this expression, and the fact that $q^*$ is the occupancy of the optimal policy $\pi^*$ under the true transition function, it follows that

$$\sum_{x,a} \sum_{h=1}^{H} r(x,a) q_h^*(x,a) = \mathbb{E}\left[ \sum_{h=1}^{H} r(X_h, \pi^*(X_h)) \Big| X_1 = x_1 \right] = V_1^*(x_1)$$

Hence, when $P \in \mathcal{P}$,

$$V_1^*(x_1) \le \sum_{x,a} \sum_{h=1}^{H} r(x,a) q_h^+(x,a) = V_1^+(x_1).$$

for the initial state $x_1$, where we have used the fact that the value of the optimal objective functions are equal due to strong duality (Lemma 10).

In order to prove the result for $x \ne x_1$ and $h \ne 1$, we consider modified linear programs defined by starting the problem at stage $h$ with all prior mass in state $x$. In this case, define the initial state as $x_h = x$, the we write the modified primal optimization problem as

$$\underset{q \in \mathcal{Q}(x), \widetilde{P} \in \Delta}{\text{maximize}} \sum_{l=h}^{H} \sum_{x,a} q_l x, a) r(x,a) \qquad (23)$$

$$\text{Subject to } \sum_{a \in \mathcal{A}} q_l(x,a) = \sum_{x' \in \mathcal{S}, a' \in \mathcal{A}} \widetilde{P}_l(x|x'a') q_{l-1}(x',a') \qquad \forall x \in \mathcal{S}, l = h+1, \dots, H$$

$$D\left( \widetilde{P}_l(\cdot|x,a), \widehat{P}_l(\cdot|x,a) \right) \le \epsilon_l(x,a), \qquad \forall (x,a) \in \mathcal{S}, l = h+1, \dots, H$$

where $\mathcal{Q}(x)$ has been modified to account for the new initial state. Observe that this problem is analogous to the primal optimization problem in (5), and hence we can apply the same techniques as used to prove Proposition 1 to show that the dual can be written as

$$\underset{V}{\text{minimize}} \ V_h(x) \qquad (24)$$

$$\text{subject to } V_l(x) \ge r(x,a) + \text{CB}_l(x,a) + \sum_{y \in \mathcal{S}} \widehat{P}_l(y|x,a) V_{l+1}(y) \quad \forall (x,a) \in \mathcal{S} \times \mathcal{A}, l \in [h:H].$$

where $\text{CB}_l(x,a) = D_*(V_{l+1}|\epsilon_l(x,a), \widehat{P}_l(\cdot|x,a))$. Analyzing this dual shows that for $l = h, \dots, H$ and $x \in \mathcal{S}$, the constraints on $V_h(x)$ here are the same as those in the full dual in (6). This means that the dual in (6) can be broken down per stage and the optimal solution can be found by a dynamic programming style algorithm. In particular, the optimal solution $V_h^+(x)$ in the complete dual in (6) is given by the optimal value of the objective function in the optimization problem in (24). Note that strong duality also applies in this modified problem since the technique used to prove this in Lemma 10 also applies here. We therefore know that $V_h^+(x) = \sum_{x,a} \sum_{l=h}^{H} \tilde{q}_l^+(x,a) r(x,a)$ where $\tilde{q}^+$ is the optimal solution to the modified LP in (23). On the event that $P$ is in the confidence set, the occupancy measure $\tilde{q}^*$ defined by the optimal policy $\pi^*$ and $P$ starting from state $x$ in stage $h$ must be a feasible solution to the LP in (23). Consequently, by the same argument as before,

$$V_h^*(x) = \sum_{x,a} \sum_{l=h}^{H} r(x,a) \tilde{q}_l^*(x,a) \le \sum_{x,a} \sum_{l=h}^{H} r(x,a) \tilde{q}_l^+(x,a) = V_h^+(x),$$

thus proving the first inequality in the statement of the proposition for all $(x,a) \in \mathcal{Z}, h = 1, \dots, H$.

We now show that $V_h^+(x) \le H - h + 1$ for all $x \in \mathcal{S}, h \in [H]$. The proof is similar to the previous case and again relies on building a new MDP from each state $x$ in stage $h$ and considering the dual. In particular, for any $x \in \mathcal{S}, h \in [H]$, in the dual LP in (24), we see that that the optimal solution to the objective function has value $V_h^+(x)$. By strong duality, this must have the same value as $\sum_{x,a} \sum_{l=h}^{H} \tilde{q}_l^+(x,a) r(x,a)$, the optimal value of the objective function of the primal optimization problem in (23) started at $x$ in stage $h$. The optimal solution $\tilde{q}^+$ must be a valid occupancy measure since by the primal constraints $q_l(x,a) \ge 0$ and $\sum_{x,a} q_h(x,a) = 1$ are satisfied. It also follows that $\sum_{x,a} q_l(x,a) = 1$ for all $l = h+1, \dots, H$ by Lemma 12. From

this it follows that $\tilde{q}_l^+(x,a) \leq 1, \forall(x,a) \in \mathcal{Z}, l = h, \ldots, H$ so combining this with the fact that $r(x,a) \in [0,1] \,\forall(x,a) \in \mathcal{Z}$, it must be the case that $\sum_{x,a} \sum_{l=h}^{H} \tilde{q}_l^+(a,x)r(x,a) \leq H - h + 1$, and so $V_{h,t}^+(x) \leq H - h + 1$ and the result holds. $\qquad\square$

**Lemma 12.** *For any feasible solution $q$ to the primal problem in* (5)*, it must hold that* $\sum_{x,a} q_h(x,a) = 1$ *for all* $h \in [H]$.

*Proof.* The proof follows by induction on $h$. For the base case, when $h = 1$,

$$\sum_{x,a} q_1(x,a) = \sum_a q_1(x_1,a) = 1$$

by the constraint $\sum_a q_1(x,a) = \mathbb{I}\{x = x_1\}$ for all $x \in \mathcal{S}$. Now assume the result holds for $h$, and we prove it for $h+1$. By the flow constraint (first constraint in (5)), for any feasible $\widetilde{P} \in \mathcal{P}$,

$$\sum_{x,a} q_{h+1}(x,a) = \sum_x \left( \sum_{x',a'} \widetilde{P}_h(x|x',a')q_h(x',a') \right) = \sum_{x',a'} q_h(x',a') = 1$$

since $\sum_x \widetilde{P}_h(x|x',a') = 1$. Thus the result holds for all $h = 1, \ldots, H$. $\qquad\square$

### A.3 Regret Bounds

In this section, we bound the regret of any algorithm that fits into our framework.

**Theorem 4.** *On the event $\cap_{t=1}^K \{P \in \mathcal{P}_t\}$, the regret is bounded with probability at least $1 - \delta$ as*

$$\mathfrak{R}_T \leq \sum_{t=1}^K \sum_{h=1}^H \left( \mathrm{CB}_{h,t}(x_{h,t}, \pi_{h,t}(x_{h,t})) + \mathrm{CB}_{h,t}^-(x_{h,t}, \pi_t(x_{h,t})) \right) + H\sqrt{2T\log(1/\delta)}$$

*where $\mathrm{CB}_{h,t}^-(x,a) = D_*(-V_{h+1,t}^+|\epsilon_{h,t}(x,a), \widehat{P}_{h,t})$ and $\mathrm{CB}_{h,t}(x,a) = D_*(V_{h+1,t}^+|\epsilon_{h,t}(x,a), \widehat{P}_{h,t})$.*

*Proof.* The proof is similar to standard proofs of regret for episodic reinforcement learning algorithms (e.g. [6, 24]) but uses Proposition 3 to simplify the probabilistic analysis and the definition of the confidence sets to simplify the algebraic analysis. For the proof, for any $h, t$, define $\Delta_{h,t}(x_{h,t}) = V_{h,t}^+(x_{h,t}) - V_h^{\pi_t}(x_{h,t})$. Then using the optimistic result from Proposition 3, on the event $\cap_{t=1}^K \{P \in \mathcal{P}_t\}$, we can write the regret as

$$\mathfrak{R}_T = \sum_{t=1}^K (V_1^*(x_{1,t}) - V_1^{\pi_t}(x_{1,t})) \leq \sum_{t=1}^K (V_{1,t}^+(x_{1,t}) - V_{1,t}^{\pi_t}(x_{1,t})) = \sum_{t=1}^K \Delta_{1,t}(x_{1,t}).$$

Then, for a fixed $h, t$, we consider $\Delta_{h,t}(x_{h,t})$ and show that this can be bounded in terms of $\Delta_{h+1,t}(x_{h+1,t})$, some confidence terms and some martingales. In particular, using the Bellman equations and the dynamic programming formulation, we can write

$\Delta_{h,t}(x_{h,t}) = V_{h,t}^+(x_{h,t}) - V_h^{\pi_t}(x_{h,t})$

$= \langle \widehat{P}_{h,t}(\cdot|x_{h,t}, a_{h,t}), V_{h+1,t}^+ \rangle + r(x_{h,t}, a_{h,t}) + \mathrm{CB}_{h,t}(x_{h,t}, a_{h,t}) - \langle P_h(\cdot|x_{h,t}, a_{h,t}), V_{h+1}^{\pi_t} \rangle - r(x_{h,t}, a_{h,t})$

$= \langle \widehat{P}_{h,t}(\cdot|x_{h,t}, a_{h,t}), V_{h+1,t}^+ \rangle - \langle P_h(\cdot|x_{h,t}, a_{h,t}), V_{h+1}^{\pi_t} \rangle + \mathrm{CB}_{h,t}(x_{h,t}, a_{h,t})$

$= \Delta_{h+1,t}(x_{h+1,t}) + \langle \widehat{P}_{h,t}(\cdot|x_{h,t}, a_{h,t}), V_{h+1,t}^+ \rangle - V_{h+1,t}^+(x_{h+1,t})$

$\qquad\qquad + V_{h+1}^{\pi_t}(x_{h+1,t}) - \langle P_h(\cdot|x_{h,t}, a_{h,t}), V_{h+1}^{\pi_t} \rangle + \mathrm{CB}_{h,t}(x_{h,t}, a_{h,t})$

$= \Delta_{h+1,t}(x_{h+1,t}) + \langle \widehat{P}_{h,t}(\cdot|x_{h,t}, a_{h,t}) - P_h(\cdot|x_{h,t}, a_{h,t}), V_{h+1,t}^+ \rangle + \zeta_{h+1,t}^{\pi} + \mathrm{CB}_{h,t}(x_{h,t}, a_{h,t})$

where in the last equality, $\zeta_{h+1,t}^{\pi}$ is a martingale difference sequence defined by

$$\zeta_{h+1,t}^{\pi} = \langle P_h(\cdot|x_{h,t}, a_{h,t}), V_{h+1,t}^+ - V_{h+1}^{\pi_t} \rangle - \left( V_{h+1,t}^+(x_{h+1,t}) - V_{h+1}^{\pi_t}(x_{h+1,t}) \right).$$

Then observe that on the event $P \in \mathcal{P}_t$,

$$\langle \widehat{P}_{h,t}(\cdot|x_{h,t}, a_{h,t}) - P_h(\cdot|x_{h,t}, a_{h,t}), V^+_{h+1,t} \rangle$$

$$\leq \max_{\widetilde{P} \in \mathcal{P}_h(x_{h,t}, a_{h,t})} \langle \widehat{P}_{h,t}(\cdot|x_{h,t}, a_{h,t}) - \widetilde{P}_h(\cdot|x_{h,t}, a_{h,t}), V^+_{h+1,t} \rangle$$

$$\leq \max_{\widetilde{P} \in \Delta} \left\{ \langle \widehat{P}_{h,t}(\cdot|x_{h,t}, a_{h,t}) - \widetilde{P}_h(\cdot|x_{h,t}, a_{h,t}), V^+_{h+1,t} \rangle : \right.$$
$$\left. D(\widetilde{P}_h(\cdot|x_{h,t}, a_{h,t}), \widehat{P}_{h,t}(\cdot|x_{h,t}, a_{h,t})) \leq \epsilon_{h,t}(x_{h,t}, a_{h,t}) \right\}$$

$$= \max_{\widetilde{P} \in \Delta} \left\{ \langle \widetilde{P}_h(\cdot|x_{h,t}, a_{h,t}) - \widehat{P}_{h,t}(\cdot|x_{h,t}, a_{h,t}), -V^+_{h+1,t} \rangle : \right.$$
$$\left. D(\widetilde{P}_h(\cdot|x_{h,t}, a_{h,t}), \widehat{P}_{h,t}(\cdot|x_{h,t}, a_{h,t})) \leq \epsilon_{h,t}(x_{h,t}, a_{h,t}) \right\}$$

$$= D_*(-V^+_{h+1,t}|\epsilon_{h,t}(x_{h,t}, a_{ht}), \widehat{P}_t(\cdot|x_{h,t}, a_{h,t}))$$

$$= \mathrm{CB}^-_{h,t}(x_{h,t}, a_{h,t})$$

This gives a recursive expression for $\Delta_{h,t}(x_{h,t})$,

$$\Delta_{h,t}(x_{h,t}) \leq \Delta_{h+1,t}(x_{h+1,t}) + \zeta^\pi_{h+1,t} + \mathrm{CB}_{h,t}(x_{h,t}, a_{h,t}) + \mathrm{CB}^-_{h,t}(x_{h,t}, a_{h,t})$$

Recursing over $h = 1, \ldots, H$, we see that,

$$\Delta_{1,t}(x_{1,t}) \leq \sum_{h=1}^H \mathrm{CB}_{h,t}(x_{h,t}, \pi_t(x_{h,t})) + \sum_{h=1}^H \mathrm{CB}^-_{h,t}(x_{h,t}, \pi_t(x_{h,t})) + \sum_{h=1}^H \zeta^\pi_{h+1,t}$$

since $\Delta_{H+1,t}(x) = 0$.

By Azuma-Hoeffdings inequality, it follows that

$$\sum_{t=1}^K \sum_{h=1}^H \zeta^\pi_{h+1,t} \leq H\sqrt{2T\log(1/\delta)}$$

with probability greater than $1 - \delta$, since the sequence has increments bounded in $[-H, H]$.

Consequently, with probability greater than $1 - \delta$, we can bound the regret by,

$$\mathfrak{R}_T \leq \sum_{t=1}^K \sum_{h=1}^H \mathrm{CB}_{h,t}(x_{h,t}, \pi_t(x_{h,t})) + + \sum_{t=1}^K \sum_{h=1}^H \mathrm{CB}^-_{h,t}(x_{h,t}, \pi_t(x_{h,t})) + H\sqrt{2T\log(1/\delta)}$$

thus giving the result. $\qquad \square$

### A.4 Upper bounding the exploration bonus

We now prove the regret bound, when we use an upper bound $D_*^\dagger$ on the conjugate $D_*$. We first need the below result that shows that the optimistic value function $V^\dagger$ in equation 9 is indeed optimistic.

**Lemma 13.** *On the event $P \in \mathcal{P}$, it holds that $V_1^*(x_1) \leq V_1^\dagger(x_1)$.*

*Proof.* We consider the dual optimization problem,

$$\operatorname*{minimize}_V \; V_1(x_1)$$

$$\text{subject to} \quad V_h(x) \geq r(x, a) + \sum_y \widehat{P}_h(y|x, a)V_{h+1}(y) + D_*\left(V_{h+1}\Big|\epsilon_h(x, a), \widehat{P}_h(\cdot|x, a)\right) \quad (25)$$

$$V_h(x) \leq H - h + 1 \quad (26)$$

which is the dual from Proposition 1, where we have added the additional constraint (26). Note that adding this additional constraint will not effect the value of the optimal solution since by Proposition 3, we know that $V^+(x) \leq H - h + 1$ for all $h = 1, \ldots, H, x \in \mathcal{S}$.

By definition of $D_*^\dagger$, it follows that for any $V_{h+1}$,

$$r(x,a) + \sum_y \widehat{P}_h(y|x,a)V_{h+1}(y) + D_*\left(V_{h+1}\Big|\epsilon_h(x,a), \widehat{P}_h(\cdot|x,a)\right)$$

$$\leq \min\left\{H - h + 1, r(x,a) + \sum_y \widehat{P}_h(y|x,a)V_{h+1}(y) + D_*^\dagger\left(V_{h+1}\Big|\epsilon_h'(x,a), \widehat{P}_h(\cdot|x,a)\right)\right\}$$

since all the original feasible solutions in stage $h + 1$ must satisfy $V_{h+1}(x) \leq H - h$. Therefore, we can replace the constraint in (25) by

$$V_h(x) \geq \min\left\{H - h + 1, r(x,a) + \sum_y \widehat{P}(y|x,a)V_{h+1}(y) + D_*^\dagger\left(V_{h+1}\Big|\epsilon_h'(x,a), \widehat{P}(\cdot|x,a)\right)\right\}$$

knowing that this will only increase the optimal value of the objective function. Since we know that by Proposition 3, that the optimal solution to the original dual optimization problem satisfies $V_1^*(x_1) \leq V_1^+(x_1)$ on the event $P \in \mathcal{P}$, it must also be the case that $V_1^*(x_{1,t}) \leq V_1^\dagger(x_{1,t})$ for $V_1^\dagger(x_{1,t})$ the optimal solution of the modified dual. Note also that the solution to the modified dual problem will take the form given in (9) by an argument similar to Lemma 11. $\square$

**Theorem 5.** *Let $D_*^\dagger(f|\epsilon', \widehat{P})$ be an upper bound on $D_*(f|\epsilon, \widehat{P})$ and $D_*(-f|\epsilon, \widehat{P})$ for every $f: \mathcal{S} \to [0, H]$, and, $\mathrm{CB}_{h,t}^\dagger(x,a) = D_*^\dagger(V_{h+1,t}^\dagger|\epsilon_{h,t}'(x,a), \widehat{P}_{h,t})$. Then, on the event $\cap_{t=1}^K \{P \in \mathcal{P}_t\}$, with probability greater than $1 - \delta$, the policy returned by the procedure in (9) incurs regret*

$$\mathfrak{R}_T \leq 2\sum_{t=1}^K \sum_{h=1}^H \mathrm{CB}_{h,t}^\dagger(x_{h,t}, \pi_{h,t}(x_{h,t})) + 4H\sqrt{2T\log(1/\delta)}.$$

*Proof.* Given the result in Lemma 13, we know that $V^\dagger$ is optimistic so the proof proceeds similarly to the case where $\mathrm{CB}_{h,t}(x,a)$ is computed exactly. In particular, let $\Delta_{h,t}^\dagger(x_{h,t}) = V^\dagger(x_{h,t}) - V^\pi(x_{h,t})$, then,

$$\mathfrak{R}_T = \sum_{t=1}^K (V_1^*(x_{1,t}) - V_1^{\pi_t}(x_{1,t})) \leq \sum_{t=1}^K (V_{1,t}^\dagger(x_{1,t}) - V_{1,t}^{\pi_t}(x_{1,t})) = \leq \sum_{t=1}^K \Delta_{h,t}^\dagger(x_{h,t})$$

and, observe that by the same argument as Theorem 4,

$$\Delta_{h,t}^\dagger(x_{h,t}) = \Delta_{h+1,t}^\dagger(x_{h+1,t}) + \left\langle \widehat{P}(\cdot|x_{h,t}, a_{h,t}) - P(\cdot|x_{h,t}, a_{h,t}), V_{h+1,t}^\dagger\right\rangle + \zeta_{h+1,t}^\dagger + \mathrm{CB}_{h,t}^\dagger(x_{h,t}, a_{h,t})$$

where $\zeta_{h+1,t}^\dagger$ is the martingale difference sequence $\zeta_{h+1,t}^\dagger = \left\langle P(\cdot|x_{h,t}, a_{h,t}, V_{h+1,t}^\dagger - V_{h+1}^{\pi_t}\right\rangle - (V_{h+1,t}^\dagger(x_{h+1,t}) - V_{h+1}^{\pi_t}(x_{h+1,t}))$. Then, on the event $P \in \mathcal{P}$,

$$\left\langle \widehat{P}_t(\cdot|x_{h,t}, a_{h,t}) - P(\cdot|x_{h,t}, a_{h,t}), V_{h+1,t}^\dagger\right\rangle$$

$$\leq \max_{\widetilde{P} \in \Delta}\left\{\left\langle \widehat{P}_t(\cdot|x_{h,t}, a_{h,t}) - \widetilde{P}, V_{h+1,t}^\dagger\right\rangle : D(\widetilde{P}, \widehat{P}_t(\cdot|x_{h,t}, a_{h,t})) \leq \epsilon_{h,t}(x_{h,t}, a_{h,t})\right\}$$

$$= \max_{\widetilde{P} \in \Delta}\left\{\left\langle \widetilde{P} - \widehat{P}_t(\cdot|x_{h,t}, a_{h,t}), -V_{h+1,t}^\dagger\right\rangle : D(\widetilde{P}, \widehat{P}_t(\cdot|x_{h,t}, a_{h,t})) \leq \epsilon(x_{h,t}, a_{h,t})\right\}$$

$$= D_*(-V_{h+1,t}^\dagger|\widehat{P}_t(\cdot|x_{h,t}, a_{h,t}), \epsilon_{h,t}(x_{h,t}, a_{ht}))$$

$$\leq D_*^\dagger(V_{h+1,t}^\dagger|\widehat{P}_t(\cdot|x_{h,t}, a_{h,t}), \epsilon_{h,t}'(x_{h,t}, a_{ht})) \leq \mathrm{CB}_{h,t}^\dagger(x_{h,t}, a_{h,t})$$

by definition of the upper bound $\mathrm{CB}_{h,t}^\dagger(x,a)$.

Using this, we can recurse over $h = 1, \ldots, H$ to get,

$$\Delta_{1,t}^\dagger(x_{1,t}) \leq 2\sum_{h=1}^H \mathrm{CB}_{h,t}^\dagger(x_{h,t}, a_{h,t}) + \sum_{h=1}^H \zeta_{h+1,t}^\dagger$$

so summing this over all episodes $t = 1, \ldots, K$ and using Azuma's inequality to bound the sum of the martingales gives the result. $\square$

## A.5 Further Details of Examples

Here we present additional results and explanations to show that many algorithms fit into our framework. The main purpose of this section is to demonstrate the use of our general results for constructing confidence sets and calculating the corresponding exploration bonuses, as well as bounding the regret. We do not aim to improve over state-of-the-art results or obtain tight constants, but we do note that several of the exploration bonuses we derive are data-dependent in a way that may possibly enable tight problem-dependent regret bounds. We refer to the works of Dann et al. [17], Zanette and Brunskill [52], Simchowitz and Jamieson [43] that demonstrate the power of data-dependent exploration bonuses for achieving such guarantees.

In several calculations below, we will use the following simple result to bound the sum of the exploration bonuses:

$$
\sum_{t=1}^{K}\sum_{h=1}^{H}\sqrt{\frac{1}{N_{h,t}(x_{h,t},a_{h,t})}} = \sum_{x\in\mathcal{S},a\in\mathcal{A}}\sum_{t=1}^{K}\sum_{h=1}^{H}\mathbb{I}\{x_{h,t}=x,a_{h,t}=a\}\sqrt{\frac{1}{N_{h,t}(x_{h,t},a_{h,t})}}
$$

$$
= \sum_{x\in\mathcal{S},a\in\mathcal{A}}\sum_{h=1}^{H}\sum_{n=1}^{N_{h,K}(x,a)}\sqrt{\frac{1}{n}} \le \sum_{x\in\mathcal{S},a\in\mathcal{A}}\sum_{h=1}^{H}2\sqrt{N_{h,K}(x,a)}
$$

$$
\le 2\sqrt{HSAT} \tag{27}
$$

where the last inequality follows due to the Cauchy–Schwarz inequality and the fact that $\sum_{x\in\mathcal{S},a\in\mathcal{A},h\in[H]} N_{h,K}(x,a) = HK = T$. We also use the modified empirical transition probability defined for any states $x, x' \in \mathcal{S}$, action $a \in \mathcal{A}$, stage $h \in [H]$ and episode $t \in [K]$ as

$$
\widehat{P}_{h,t}^{+}(x'|x,a) = \frac{\max\{1, N_{h,t}(x,a,x')\}}{N_{h,t}(x,a)} \tag{28}
$$

and note that this only differs from $\widehat{P}_{h,t}(x'|x,a)$ if $N_{h,t}(x,a,x') = 0$. Consequently,

$$
|\widehat{P}_{h,t}^{+}(x'|x,a) - \widehat{P}_{h,t}(x'|x,a)| = \left|\frac{\max\{1, N_{h,t}(x,a,x')\}}{N_{h,t}(x,a)} - \frac{N_{h,t}(x,a,x')}{N_{h,t}(x,a)}\right| \le \frac{1}{N_{h,t}(x,a)} \tag{29}
$$

In several cases, we define the primal confidence sets using $\widehat{P}^{+}$ as the reference model rather than $\widehat{P}$ to avoid division by 0. Note that doing this results in dual formulations that involve $\widehat{P}^{+}$ rather than $\widehat{P}$. However, since we are still optimizing over the space of probability distributions in the primal, it holds that the optimal value of the dual objective will still be bounded by $H$. We can also use Equation (29) to bound the empirical variance of any function $z : \mathcal{S} \to [0, H]$ under $\widehat{P}^{+}$,

$$
\widehat{\mathbb{V}}^{+}(z) = \sum_{y}\widehat{P}^{+}(y)(z(y) - \langle\widehat{P}^{+}, z\rangle)^2 \le \sum_{y}\widehat{P}^{+}(y)\left(2(z(y) - \langle\widehat{P}, z\rangle)^2 + 2(\frac{HS}{N})^2\right)
$$

$$
\le 2\sum_{y}\widehat{P}(y)(z(y) - \langle\widehat{P}, z\rangle)^2 + \frac{2HS + 2(\frac{HS}{N})^2}{N} + \frac{H^2S^2}{N^2} \le 2\widehat{\mathbb{V}}(z) + \frac{2HS}{N} + \frac{3H^2S^2}{N^2}. \tag{30}
$$

### A.5.1 Total variation distance

We start with the classic choice of the $\ell_1$ distance $D(p, p') = \|p - p'\|_1$ which underlies the seminal UCRL2 algorithm of Jaksch et al. [24]. Defining the confidence sets used in episode $t$ as

$$
\mathcal{P}_t = \left\{\widetilde{P} \in \Delta : \left\|\widetilde{P}_h(\cdot|x,a) - \widehat{P}_t(\cdot|x,a)\right\|_1 \le \epsilon_{h,t}(x,a) \quad \forall(x,a) \in \mathcal{Z}, h \in [H]\right\}
$$

for $\quad \epsilon_{h,t}(x,a) = \sqrt{\frac{2S\log(2SAT/\delta)}{N_{h,t}(x,a)}}$

we know that $P \in \mathcal{P}_t$ for all $t = 1, \ldots, K$ with probability greater than $1 - \delta$ [24]. Then, the conjugate distance is,

$$D_*(f|\epsilon, \widehat{P}) = \max_{P \in \Delta} \left\{ \langle P - \widehat{P}, f \rangle \, \Big| \, \left\| P - \widehat{P} \right\|_1 \le \epsilon \right\} = \min_{\lambda \in \mathbb{R}} \max_{P \ge 0} \left\{ \langle P - \widehat{P}, f - \lambda \mathbf{1} \rangle \, \Big| \, \left\| P - \widehat{P} \right\|_1 \le \epsilon \right\}$$

$$\le \min_{\lambda \in \mathbb{R}} \max_{P \in \mathbb{R}^S} \left\{ \langle P - \widehat{P}, f - \lambda \mathbf{1} \rangle \, \Big| \, \left\| P - \widehat{P} \right\|_1 \le \epsilon \right\} \le \epsilon \min_{\lambda \in \mathbb{R}} \| f - \lambda \mathbf{1} \|_\infty \le \epsilon \, sp(f)/2$$

where we have defined $\lambda$ as the Lagrange multiplier of the constraint $\sum_x P(x) = 1 = \sum_x \widehat{P}(x)$, used the fact that the dual norm of the $\ell_1$ norm is the $\ell_\infty$ norm and, denoted by $sp(f) = \max_x f(x) - \min_x f(x)$ the span of $f$. Noting that a similar result holds for $D_*(-f|\epsilon, \widehat{P})$, we can define $D_*^\dagger(f|\epsilon, \widehat{P}) = \epsilon \, sp(f)/2$, and use the exploration bonus

$$\mathrm{CB}_{h,t}^\dagger(x, a) = \epsilon_{h,t}(x, a) sp(V_{h+1,t}^\dagger)/2$$

Since we are clipping $V_h^+$ to be in the range $[0, H - h + 1]$, we can bound $sp(V_h^\dagger) \le H$. Applying Theorem 5 and using the bound of Equation (27) to bound the sum of the exploration bonuses shows that the regret of this algorithm is bounded by $\widetilde{O}(S\sqrt{AH^3T})$. This recovers the classic UCRL2 guarantees that can be deduced from the work of [24].

### A.5.2 Variance-weighted $\ell_\infty$ norm

We can get tighter bounds by using the empirical Bernstein inequality [35] to constrain the transition function. Here, we use $\widehat{P}^+ = \frac{\max\{1, N_{h,t}(x,a,y)\}}{N_{h,t}(x,a)}$ as the reference model in the primal confidence sets. The constraints considered here are related to those used in the UCRL2B algorithm of Fruit et al. [21]. Specifically, we can apply the empirical Bernstein inequality to show that the following bound holds for all $x, a, x', h, t$ with probability at least $1 - \delta$:

$$\left| \widehat{P}_{h,t}^+(x'|x,a) - P_h(x'|x,a) \right| \le \left| \widehat{P}_{h,t}(x'|x,a) - P_h(x'|x,a) \right| + \left| \widehat{P}_{h,t}^+(x'|x,a) - \widehat{P}_{h,t}(x'|x,a) \right|$$

$$\le \sqrt{\frac{2\widehat{P}_{h,t}(x'|x,a)\left(1 - \widehat{P}_h(x'|x,a)\right)\log(HS^2AT/\delta)}{N_{h,t}(x,a)}} + \frac{7\log(HS^2AT/\delta)}{3N_{h,t}(x,a)} + \frac{1}{N_{h,t}(x,a)}$$

$$\le \sqrt{\frac{2\widehat{P}_{h,t}(x'|x,a)\log(HS^2AT/\delta)}{N_{h,t}(x,a)}} + \frac{7\log(HS^2AT/\delta)}{3N_{h,t}(x,a)} + \frac{1}{N_{h,t}(x,a)}$$

$$\le \sqrt{\frac{2\widehat{P}_{h,t}^+(x'|x,a)\log(HS^2AT/\delta)}{N_{h,t}(x,a)}} + \frac{7\log(HS^2AT/\delta)}{3N_{h,t}(x,a)} + \frac{1}{N_{h,t}(x,a)}$$

$$\le 6\log(HS^2AT/\delta)\sqrt{\frac{\widehat{P}_{h,t}^+(x'|x,a)}{N_{h,t}(x,a)}}$$

The last inequality follows from the definition of the reference model that guarantees that $N_{h,t}(x,a)\widehat{P}_{h,t}^+(y|x,a) = \max\{N_{h,t}(x,a,y), 1\} \ge 1$.

In what follows, we will state a confidence set inspired by the above result using the divergence measure $D(P, \widehat{P}^+) = \max_x \frac{(P(x) - \widehat{P}^+(x))^2}{\widehat{P}^+(x)}$, which is easily seen to be positive homogeneous and convex in both $P$ and $\widehat{P}^+$. Defining $\epsilon_{h,t}(x,a) = \frac{36\log^2(HS^2AT/\delta)}{N_{h,t}(x,a)}$, we define the confidence sets used in episode $t$ as

$$\mathcal{P}_h(\cdot|x,a) = \left\{ P_h(\cdot|x,a) \in \Delta : \max_y \frac{(\widetilde{P}(y|x,a) - \widehat{P}_{h,t}^+(y|x,a))^2}{\widehat{P}_{h,t}^+(y|x,a)} \le \epsilon_{h,t}(x,a) \right\}$$

and $\mathcal{P} = \cap_{x,a,h} \{\mathcal{P}_h(\cdot|x,a)\}$. By the above argument, we know that $P \in \mathcal{P}$ with probability greater than $1 - \delta$.

The corresponding conjugate distance can be expressed by defining $\lambda$ as the Lagrange multiplier of the constraint $\sum_x P(x) = 1$ and writing

$$D_*(f|\epsilon, \widehat{P}^+) = \max_{P \in \Delta} \left\{ \langle P - \widehat{P}^+, f \rangle \,\middle|\, \max_{x \in \mathcal{S}} \frac{(P(x) - \widehat{P}^+(x))^2}{\widehat{P}^+(x)} \leq \epsilon \right\}$$

$$= \min_{\lambda \in \mathbb{R}} \max_{P \geq 0} \left\{ \langle P - \widehat{P}^+, f - \lambda \mathbf{1} \rangle - \lambda \left( \sum_x \widehat{P}^+(x) - 1 \right) \,\middle|\, \max_{x \in \mathcal{S}} \frac{|P(x) - \widehat{P}(x)|}{\sqrt{\widehat{P}^+(x)}} \leq \sqrt{\epsilon} \right\}$$

$$\leq \min_{|\lambda| \leq H + \frac{SH}{N}} \max_{P \in \mathbb{R}^S} \left\{ \langle P - \widehat{P}^+, f - \lambda \mathbf{1} \rangle - \lambda \sum_x (\widehat{P}^+(x) - \widehat{P}(x)) \,\middle|\, \max_{x \in \mathcal{S}} \frac{|P(x) - \widehat{P}(x)|}{\sqrt{\widehat{P}^+(x)}} \leq \sqrt{\epsilon} \right\}$$

$$\leq \min_{|\lambda| \leq H + \frac{SH}{N}} \sum_x \left| (f(x) - \lambda)\sqrt{\widehat{P}^+(x)} \right| \sqrt{\epsilon} + \left( H + \frac{SH}{N} \right) \frac{1}{N}$$

$$\leq \sqrt{\epsilon} \sum_x \sqrt{\widehat{P}^+(x)} |f(x) - \widehat{P}^+ f| + \frac{2SH}{N}.$$

The same technique can be used to bound $D_*(-f|\epsilon, \widehat{P})$, so we can define $D_*^\dagger(f|\epsilon, \widehat{P}) = \sqrt{\epsilon} \sum_x \sqrt{\widehat{P}^+(x)} |f(x) - \widehat{P}^+ f| + \frac{2SH}{N}$ and write the inflated exploration bonus in the form

$$\mathrm{CB}_{h,t}^\dagger(x, a) = \sqrt{\epsilon_{h,t}(x,a)} \sum_y \sqrt{\widehat{P}_{h,t}^+(y|x,a)} |V_{h+1}^\dagger(y) - \widehat{P}_{h,t}^+ V_{h+1}^\dagger| + \frac{2SH}{N_{h,t}(x,a)}.$$

By Theorem 5, we know that in order to bound the regret of this algorithm, we need to be able to bound the sum of these exploration bonuses. For this, note that by the Cauchy–Schwarz inequality, and a similar argument to (30),

$$\sqrt{\epsilon_{h,t}(x,a)} \sum_y \sqrt{\widehat{P}_{h,t}^+(y|x,a)} |V_{h+1}^\dagger(y) - \widehat{P}_{h,t}^+ V_{h+1}^\dagger|$$

$$\leq \sqrt{\epsilon_{h,t}(x,a)} \sum_y \sqrt{\widehat{P}_{h,t}(y|x,a)} |V_{h+1}^\dagger(y) - \widehat{P}_{h,t} V_{h+1}^\dagger| + SH \sqrt{\frac{\epsilon_{h,t}(x,a)}{N_{h,t}(x,a)}} \left( 2 + \sqrt{\frac{H}{SN_{h,t}(x,a)}} \right)$$

$$= \sqrt{\epsilon_{h,t}(x,a)} \sum_{y:P(y)>0} \sqrt{\widehat{P}_{h,t}(y|x,a)} |V_{h+1}^\dagger(y) - \widehat{P}_{h,t} V_{h+1}^\dagger| + 3SH \sqrt{\frac{\epsilon_{h,t}(x,a)}{N_{h,t}(x,a)}}$$

$$\leq \sqrt{\epsilon_{h,t}(x,a)} \sqrt{\Gamma_h(x,a) \sum_{y:P(y)>0} \widehat{P}_{h,t}^+(y|x,a)(V_{h+1}^\dagger(y) - \widehat{P}_{h,t} V_{h+1}^\dagger)^2} + 3SH \sqrt{\frac{\epsilon_{h,t}(x,a)}{N_{h,t}(x,a)}}$$

$$\leq \sqrt{\epsilon_{h,t}(x,a) \Gamma \widehat{\mathbb{V}}_{h,t}(V_{h+1}^\dagger)} + 3SH \sqrt{\frac{\epsilon_{h,t}(x,a)}{N_{h,t}(x,a)}}$$

where $\Gamma_h(x,a)$ is the number of next states which can be reached from state $x$ after playing action $a$ in stage $h$ with positive probability, and $\Gamma$ is a uniform upper bound on $\Gamma_h(x,a)$ that holds for all $x, a$, and $\widehat{\mathbb{V}}_{h,t}$ is the empirical variance using all data from stage $h$ up to episode $t$. In order to bound $\mathrm{CB}_{h,t}^\dagger(x_{h,t}, a_{h,t}) \leq \sum_{t=1}^K \sum_{h=1}^H (\sqrt{\epsilon_{h,t}(x_{h,t}, a_{h,t}) \Gamma \widehat{\mathbb{V}}_{h,t}(V_{h+1,t}^\dagger)} + 3SH \sqrt{\frac{\epsilon_{h,t}(x,a)}{N_{h,t}(x,a)}})$, we use the Cauchy–Schwarz inequality and techniques similar to Lemma 10 in [6] or Lemma 5 in [21] to show

that

$$\sum_{t=1}^{K}\sum_{h=1}^{H}\mathrm{CB}_{h,t}^{\dagger}(x_{h,t},a_{h,t})$$

$$\leq C_1\sqrt{\Gamma L}\sqrt{\sum_{t=1}^{K}\sum_{h=1}^{H}\frac{1}{N_{h,t}(x_{h,t},a_{h,t})}\sum_{t=1}^{K}\sum_{h=1}^{H}\widehat{\mathbb{V}}_{h,t}(V_{h+1,t}^{\dagger})}+C_4 SH\sqrt{L}\sum_{t=1}^{K}\sum_{h=1}^{H}\frac{1}{N_{h,t}(x_{h,t},a_{h,t})}$$

$$\leq C_1\sqrt{\Gamma L}\sqrt{SA\log(T)\left(\sum_{t=1}^{K}\sum_{h=1}^{H}\mathbb{V}_h(V_{h+1}^{\pi_t})+C_2 H^2\sqrt{T\log(T)}\right)}+C_4 SH\sqrt{L}SA\log(T)$$

$$\leq C_1\sqrt{\Gamma L}\sqrt{SA\log(T)\left(HT+C_3 H^2\sqrt{TL}+C_2 H^2\sqrt{T\log(T)}\right)}+C_4 SH\sqrt{L}SA\log(T)$$

$$=\widetilde{O}(H\sqrt{\Gamma SAT})$$

for some constants $C_1, C_2, C_3, C_4 > 0$, $L = \log(HS^2AT/\delta)$ and $\mathbb{V}_h$ the variance under $P_h$, where the penultimate inequality follows from [6] and the last inequality holds for $S^3A \leq T\Gamma$. This recovers the regret bounds of Fruit et al. [21].

### A.5.3 Relative entropy

Inspired by the KL-UCRL algorithm of Filippi et al. [20], we also consider the relative entropy (or Kullback–Leibler divergence, KL divergence) between $\widehat{P}$ and $\widetilde{P}$ as a divergence measure. The relative entropy between two discrete probability distributions $p$ and $q$ is defined as

$$D(p,q) = \sum_x p(x)\log\frac{p(x)}{q(x)},$$

provided that $p(x) = 0$ holds whenever $q(x) = 0$. Being an $f$-divergence, the KL divergence satisfies the conditions necessary for our analysis: positive homogeneous and jointly convex in its arguments $(p,q)$. However, it is not symmetric in its arguments, which suggests that it can be used for defining confidence sets in two different ways, corresponding to the ordering of $P$ and $\widehat{P}$. We describe the confidence sets and the resulting exploration bonuses below.

**Forward KL-Divergence.** We first consider constraining the divergence $D(P,\widehat{P}) = \sum_y P(y)\log\left(\frac{P(y)}{\widehat{P}(y)}\right)$. To address the issue that the empirical transition probabilities $\widehat{P}(y)$ may be zero for some $y \in \mathcal{S}$, we define the divergence with respect to $\widehat{P}^+$ (as defined in equation (28)) and use the so-called *unnormalized relative entropy* to account for the fact that $\widehat{P}^+$ may not be a valid probability distribution. Specifically, in what follows, we consider the following divergence measure:

$$D(P,\widehat{P}) = \sum_y P(y)\log\left(\frac{P(y)}{\widehat{P}^+(y)}\right) + \sum_y(\widehat{P}^+(y) - P(y)).$$

The following concentration result will be helpful for the construction of the confidence sets.

**Lemma 14.** *With probability greater than* $1 - \delta$*, it holds that for every episode* $t$*, stage* $h$ *and state-action pair* $(x,a)$*,*

$$D(P_h(x,a),\widehat{P}_{h,t}^+(x,a)) \leq \frac{18S\log(HSAT/\delta)}{N_{h,t}(x,a)}$$

*Proof.* We consider a fixed $h, t, x, a$, and for ease of notation remove the dependence of $P, \widehat{P}$ on $h, t, x, a$. With probability greater than $1 - \frac{\delta}{HTSA}$, it follows that

$$\sum_y P(y) \log\left(\frac{P(y)}{\widehat{P}^+(y)}\right) + \sum_y (\widehat{P}^+(y) - P(y)) \leq \sum_y P(y)\left(\frac{P(y)}{\widehat{P}^+(y)} - 1\right) + \sum_y (\widehat{P}^+(y) - P(y))$$
$$\text{(Since } \log(x) \leq x - 1 \text{ for } x > 0)$$

$$= \sum_y \frac{P^2(y) - P(y)\widehat{P}^+(y)}{\widehat{P}^+(y)} + \sum_y (\widehat{P}^+(y) - P(y))$$

$$= \sum_y \frac{(P(y) - \widehat{P}^+(y))^2}{\widehat{P}^+(y)}$$

$$\leq 2\sum_y \frac{(P(y) - \widehat{P}(y))^2}{\widehat{P}^+(y)} + 2\sum_y \frac{(\widehat{P}(y) - \widehat{P}^+(y))^2}{\widehat{P}^+(y)}$$

$$\leq 2\sum_y \frac{2\widehat{P}(y)\log(HS^2AT/\delta)/N + 6\log^2(HS^2AT/\delta)/N^2}{\widehat{P}^+(y)} + 2\sum_y \frac{1}{N^2\widehat{P}^+(y)}$$
$$\text{(By Bernstein's inequality and (29))}$$

$$\leq \frac{18S\log(HS^2AT/\delta)}{N}$$

where the last inequality follows since by definition $\widehat{P}^+(y)N \geq 1$. Since this holds for each $h, t, x, a$ with probability greater than $1 - \frac{\delta}{HTSA}$, by the union bound, it follows that it holds simultaneously for all $h, t, x, a$ with probability greater than $1 - \delta$. $\square$

Given the above result, we define our confidence set as

$$\mathcal{P}_{h,t}(\cdot|x,a) = \left\{\widetilde{P}_h(\cdot|x,a) \in \Delta \,\middle|\, \sum_{x'} \widetilde{P}_h(x'|x,a)\log\frac{\widetilde{P}_h(x'|x,a)}{\widehat{P}^+_{h,t}(x'|x,a)} \leq \epsilon_{h,t}(x,a)\right\}$$

$$\text{for} \qquad \epsilon_{h,t}(x,a) = \frac{CS\log(HSAT/\delta)}{N_{h,t}(x,a)}$$

for some constant $C > 0$. Using the notation $\text{KL}(p,q) = \sum_y p(y)\log(p(y)/q(y))$ to denote the normalized KL divergence, the conjugate of the above divergence can be written as

$$D_*(z|\epsilon, \widehat{P}^+) = \max_{\widetilde{P}\in\Delta}\left\{\left\langle z, \widetilde{P} - \widehat{P}^+\right\rangle \,\middle|\, D(\widetilde{P}, \widehat{P}^+) \leq \epsilon\right\}$$

$$= \min_{\lambda \geq 0}\max_{\widetilde{P}\in\Delta}\left\{\left\langle z, \widetilde{P} - \widehat{P}^+\right\rangle - \lambda\left(D(\widetilde{P}, \widehat{P}^+) - \epsilon\right)\right\}$$

$$= \min_{\lambda \geq 0}\max_{\widetilde{P}\in\Delta}\left\{\left\langle z, \widetilde{P} - \widehat{P}^+\right\rangle - \lambda\left(\text{KL}(\widetilde{P}, \widehat{P}^+) + \left\langle \mathbf{1}, \widehat{P}^+ - \widetilde{P}\right\rangle - \epsilon\right)\right\}$$

$$= \min_{\lambda \geq 0}\max_{\widetilde{P}\in\Delta}\left\{\left\langle z, \widetilde{P} - \widehat{P}^+\right\rangle - \lambda\left(\text{KL}(\widetilde{P}, \widehat{P}^+) - \epsilon'\right)\right\}$$

$$= \min_{\lambda \geq 0}\left\{\lambda\log\sum_{x'}\widehat{P}^+(x')e^{z(x)/\lambda} - \sum_{x'}\widehat{P}^+(x')z(x') + \lambda\epsilon'\right\}$$

where we defined $\epsilon' = \epsilon + 1 - \langle \mathbf{1}, \widehat{P}^+\rangle$ and used the well-known Donsker–Varadhan variatonal formula (see, e.g., [10, Corollary 4.15]) in the last line. Thus, the exploration bonus can be efficiently calculated by a line-search procedure to find the $\lambda$ minimizing the expression above.

A more tractable bound on the exploration bonus can be provided by noting that, for a vector $z$ with $\|z\|_\infty \leq H$, we have

$$D_*(z|\epsilon, \widehat{P}^+) = \min_{\lambda \geq 0} \left\{ \lambda \log \sum_y \widehat{P}^+(y) e^{z(y)/\lambda} - \sum_y \widehat{P}^+(y) z(y) + \lambda \epsilon' \right\}$$

$$= \min_{\lambda \geq 0} \left\{ \lambda \log \sum_y \widehat{P}^+(y) e^{(z(y) - \langle \widehat{P}^+, z \rangle)/\lambda} + \lambda \epsilon' \right\}$$

$$\leq \min_{\lambda \in [0, H]} \left\{ \lambda \log \sum_y \widehat{P}^+(y) e^{(z(y) - \langle \widehat{P}^+, z \rangle)/\lambda} + \lambda \epsilon' \right\}$$

$$\leq \min_{\lambda \in [0, H]} \left\{ \frac{1}{\lambda} \sum_y \widehat{P}^+(y) \left( z(y) - \langle \widehat{P}^+, z \rangle \right)^2 + \lambda \epsilon' \right\}$$

$$\leq 2 \sqrt{\epsilon' \sum_y \widehat{P}^+(y)(z(y) - \langle \widehat{P}^+, z \rangle)^2} = 2 \sqrt{\epsilon' \widehat{\mathbb{V}}^+(z)}$$

where we used the inequality $\lambda \log \mathbb{E}^+[e^{X/\lambda}] \leq \mathbb{E}^+[X] + \frac{1}{\lambda} \mathbb{E}^+[X^2]$ for $\mathbb{E}^+[X] = \sum_x \widehat{P}^+(x) x$ that holds as long as $|X| \leq \lambda$ holds almost surely, and the result in Equation (29) several times. We also use the notation $\widehat{\mathbb{V}}^+(z)$ to denote the variance of $z$ under $\widehat{P}^+$. Thus, defining

$$\epsilon'_{h,t}(x, a) = \epsilon_{h,t}(x, a) + \sum_y \widehat{P}^+(y|x, a) - 1 \leq \epsilon_{h,t}(x, a) + \frac{S - \Gamma}{N_{t,h}(x, a)} = \widetilde{O}\left( \frac{S}{N_{h,t}(x, a)} \right),$$

the exploration bonus can be bounded as $\mathrm{CB}_{h,t}(x, a) \leq 2 \sqrt{\epsilon'_{h,t}(x, a) \widehat{\mathbb{V}}^+_{h,t}(V^+_{h+1,t})}$, and using an identical argument yields the same bound for $\mathrm{CB}^-_{h,t}(x, a)$.

By (29), $\widehat{\mathbb{V}}^+(z) \leq 2 \widehat{\mathbb{V}}(z) + \frac{2HS}{N} + \frac{3H^2 S^2}{N^2}$ and so the exploration bonus can be bounded in the same way as in the case of variance-weighted $\ell_\infty$ constraints, plus some lower order terms that scale with $1/N$. The sum of these lower order terms can be straightforwardly bounded by a simple adaptation of the calculations in Equation (27). Overall, the sum of the confidence bounds can be bounded as

$$\sum_{t=1}^K \sum_{h=1}^H (\mathrm{CB}_{h,t}(x_{h,t}, a_{h,t}) + \mathrm{CB}^-_{h,t}(x_{h,t}, a_{h,t})) \leq C_1 HS\sqrt{AT} + C_2 H^2 S^2 A \log T$$

for some $C_1, C_2 = O(\log(HSAT/\delta))$. Hence the regret can be bounded by $\widetilde{O}(HS\sqrt{AT})$.

**Reverse KL-Divergence.** We now consider defining confidence sets in terms of the second argument of the KL divergence, corresponding the the original KL-UCRL algorithm proposed by Filippi et al. [20], Talebi and Maillard [48]. Specifically, define,

$$\mathcal{P}_{h,t}(\cdot|x, a) = \left\{ \widetilde{P}_h(\cdot|x, a) \in \Delta \,\middle|\, \sum_{x'} \widehat{P}_h(x'|x, a) \log \frac{\widehat{P}_h(x'|x, a)}{\widetilde{P}_{h,t}(x'|x, a)} \leq \epsilon_{h,t}(x, a) \right\}$$

for $\qquad \epsilon_{h,t}(x, a) = \dfrac{CS \log(HSAT/\delta)}{N_{h,t}(x, a)}.$

for some constant $C > 0$. As shown by Filippi et al. [20], for an appropriate choice of $C$, this confidence set is guaranteed to capture the true transition function in all episodes with probability greater than $1 - \delta$.

The conjugate of this distance for a fixed $x, a$ can be bounded as

$$D_*(z|\epsilon, \widehat{P}) = \max_{\widetilde{P} \in \Delta} \left\{ \left\langle z, \widetilde{P} - \widehat{P} \right\rangle \middle| D(\widetilde{P}, \widehat{P}) \leq \epsilon \right\}$$

$$= \min_{\lambda \geq 0} \max_{\widetilde{P} \in \Delta} \left\{ \left\langle z, \widetilde{P} - \widehat{P} \right\rangle - \lambda(D(\widetilde{P}, \widehat{P}) - \epsilon) \right\}$$

$$\leq \min_{\lambda \geq 0} \max_{\widetilde{P} \in \Delta} \left\{ \left\langle z, \widetilde{P} - \widehat{P} \right\rangle - \lambda(1/2 \|\widetilde{P} - \widehat{P}\|_1^2 - \epsilon) \right\} \qquad \text{(By Pinsker's inequality)}$$

$$\leq sp(z) \sqrt{2\epsilon}$$

where the last inequality follows by an argument similar to the results for the total variation distance in Section A.5.1 using the fact that the dual of the $\ell_1$ norm is the $\ell_\infty$ norm.

Similarly, it can be shown that $D_*(-z|\epsilon, \widehat{P}) \leq sp(z)\sqrt{2\epsilon}$. Therefore, we define the confidence bounds,

$$\mathrm{CB}_{h,t}^\dagger(x,a) = sp(V_{h+1,t}^\dagger)\sqrt{2\epsilon_{h,t}(x,a)}.$$

By Theorem 5, we know the regret can be bounded in terms of the sum of these confidence bounds. Consequently, using equation 27, we see that,

$$\sum_{t=1}^{K}\sum_{h=1}^{H} \mathrm{CB}_{h,t}^\dagger(x_{h,t}, a_{h,t}) \leq HS\sqrt{2HAT}\log(HSAT/\delta).$$

Hence the regret can be bounded by $\widetilde{O}(S\sqrt{H^3AT})$. This matches the regret bound in Filippi et al. [20]. Using an alternative analysis essentially corresponding to a tighter bound on the conjugate distance, Talebi and Maillard [48] were able to prove a regret bound of $\widetilde{O}\left(\sqrt{S\sum_{h,x,a} \mathbb{V}_{h-1}(V_h^*(x,a))T}\right)$ for KL-UCRL where $\mathbb{V}_{h-1}(V_h^*(x,a))$ is the variance of $V_h^*$ after playing action $a$ from state $s$ in stage $h-1$. We conjecture that it is possible to obtain a regret bound of $\widetilde{O}(H\sqrt{\Gamma SAT})$ by combining the techniques of Talebi and Maillard [48] and Azar et al. [6].

### A.5.4 $\chi^2$-divergence

We can also use the Pearson $\chi^2$-divergence to define the primal confidence sets in (5). Specifically, we consider the distance

$$D(P, \widehat{P}^+) = \sum_y \frac{(P(y) - \widehat{P}^+(y))^2}{\widehat{P}^+(y)},$$

for $\widehat{P}^+$ defined as in equation (28) and note that similar results hold for the distance $D(P, \widehat{P}) = \sum_y \frac{P^2(y) - \widehat{P}^2(y)}{\widehat{P}(y)}$. We will use $\widehat{P}^+$ as the reference model for the primal confidence sets. Using the empirical Bernstein inequality [35], we see that with probability greater than $1 - \delta$, for all episodes $t$, $a \in \mathcal{A}, x \in \mathcal{S}, h \in [H]$,

$$D(P_h(\cdot|x,a), \widehat{P}_{h,t}^+(\cdot|x,a)) = \sum_y \frac{(P_h(y|x,a) - \widehat{P}_{h,t}^+(y|x,a))^2}{\widehat{P}_{h,t}^+(y|x,a)}$$

$$\leq 2\sum_y \frac{(P_h(y|x,a) - \widehat{P}_{h,t}(y|x,a))^2}{\widehat{P}_{h,t}^+(y|x,a)} + 2\sum_y \frac{(\widehat{P}_h(y|x,a) - \widehat{P}_{h,t}^+(y|x,a))^2}{\widehat{P}_{h,t}^+(y|x,a)}$$

$$\leq \sum_y \left(\frac{2\widehat{P}_{h,t}(y|x,a)(1 - \widehat{P}_{h,t}(y|x,a))\log(HS^2AT/\delta)}{N_{h,t}(x,a)\widehat{P}_{h,t}^+(y|x,a)} + \frac{49\log^2(HS^2AT/\delta)}{9N_{h,t}^2(x,a)\widehat{P}_{h,t}^+(y|x,a)}\right) + \frac{2S}{N_{h,t}(x,a)}$$

$$\leq \sum_y \left(\frac{2\widehat{P}_{h,t}^+(y|x,a)\log(HS^2AT/\delta)}{N_{h,t}(x,a)\widehat{P}_{h,t}^+(y|x,a)} + \frac{49\log^2(HS^2AT/\delta)}{9N_{h,t}(x,a)}\right) + \frac{2S}{N_{h,t}(x,a)}$$

$$\leq \frac{11S\log^2(HS^2AT/\delta)}{N_{h,t}(x,a)}$$

where the second to last inequality follows since $N_{h,t}(x,a)\widehat{P}_{h,t}^+(y|x,a) = \max\{1, N_{h,t}(x,a,y)\} \geq 1$, and $\widehat{P}_{h,t}^+(y|x,a) \geq \widehat{P}_{h,t}(y|x,a)$. We can then define the confidence sets as

$$\mathcal{P}_{h,t}(\cdot|x,a) = \left\{\widetilde{P}_h \in \Delta \,\middle|\, D(\widetilde{P}_h(\cdot|x,a), \widehat{P}_{h,t}^+(\cdot|x,a)) \leq \epsilon_{h,t}(x,a)\right\} \text{ for } \epsilon_{h,t}(x,a) = \frac{11S\log^2(HS^2AT/\delta)}{N_{h,t}(x,a)}.$$

Furthermore, the conjugate $D_*(V|\epsilon, \widehat{P}^+)$ can be written as follows:

$$D_*(V|\epsilon, \widehat{P}^+) = \max_{P \in \Delta} \left\{ \langle P - \widehat{P}^+, V \rangle : D(P, \widehat{P}^+) \le \epsilon \right\}$$

$$= \min_{\lambda \in \mathbb{R}} \max_{P \ge 0} \left\{ \langle P - \widehat{P}^+, V - \lambda \mathbf{1} \rangle - \lambda (\sum_y \widehat{P}^+(y) - 1) : \left\| \frac{P - \widehat{P}^+}{\sqrt{\widehat{P}^+}} \right\|_2^2 \le \epsilon \right\}$$

$$= \min_{\lambda \in \mathbb{R}} \max_{P} \left\{ \langle P - \widehat{P}^+, V - \lambda \mathbf{1} \rangle - \lambda \sum_y (\widehat{P}^+(y) - \widehat{P}(y)) : \left\| \frac{P - \widehat{P}^+}{\sqrt{\widehat{P}^+}} \right\|_2 \le \sqrt{\epsilon} \right\}$$

$$= \min_{\lambda \le H + \frac{SH}{N}} \sqrt{\epsilon \sum_y \widehat{P}^+(y)(V(y) - \lambda)^2} + \left( H + \frac{SH}{N} \right) \frac{1}{N}$$

$$\le \sqrt{\epsilon \widehat{\mathbb{V}}^+(V)} + \frac{2SH}{N}$$

where we have used properties of the dual of the weighted $\ell_2$ norm. Therefore, both $\mathrm{CB}_{h,t}(x,a)$ and $\mathrm{CB}_{h,t}^-(x,a)$ can be upper-bounded by for $\mathrm{CB}_{h,t}^\dagger(x,a) = \sqrt{\epsilon_{h,t}(x,a)\widehat{\mathbb{V}}_{h,t}^+(V_{h+1,t}^+)}$ and we can apply Theorem 5 to show that the regret is bounded by the sum of these exploration bonuses. Following the same steps as in Section A.5.2 and using the bound on the variance under $\widehat{P}^+$ in (30), this eventually leads to a regret bound of $\widetilde{O}(HS\sqrt{AT})$.

It is interesting to note that Maillard et al. [33] considered similar confidence sets using a *reverse* $\chi^2$-divergence defined as $D(p,q) = \sum_y \frac{q^2(y) - p^2(y)}{p(y)}$. Using this distance with a feasible confidence set would fit into our framework. However, for their regret analysis, Maillard et al. [33] impose the additional constraint that for all $x'$ such that $\widetilde{P}_{h,t}(x'|x,a) > 0$, it must also hold that $\widetilde{P}_{h,t}(x'|x,a) > p_0$ for some positive $p_0$. Unfortunately, this constraint makes the set $\mathcal{P}$ non-convex[7] and thus their eventual approach does not entirely fit into our framework. Finally, we note that the bounds of Maillard et al. [33] replace a factor of $S$ appearing in our bounds by $1/p_0$, which may in an inferior bound when $p_0$ is small. Overall, we believe that the Pearson $\chi^2$-divergence we propose in this section can remove this limitation of the analysis of Maillard et al. [33] while also retaining the strong problem-dependent character of their bounds.

## B Results for Linear Function Approximation

In this section, we provide proofs of the results in the linear function approximation setting. Throughout the analysis, we will use the notation

$$C_t(\delta) = 2H\sqrt{d \log (1 + tR^2/\lambda) + \log(1/\delta)} + C_P H \sqrt{\lambda d}$$

where $C_P$ is such that $\|m_{h,a}(x)\|_1 \le C_P$ for every row $m_{h,a}(x)$ of $M_{h,a}$ and $R$ is such that $\|\varphi(x)\|_2 \le R$ for all $x \in \mathcal{S}$. We also define the event

$$\mathcal{E}_{h,a,t}(g,\delta) = \left\{ \left\| \left( M_{h,a} - \widehat{M}_{h,a,t} \right) g \right\|_{\Sigma_{h,a,t-1}} \le C_t(\delta) \right\}.$$

We start by proving our key concentration result that will be used for deriving our confidence sets.

**Proposition 15.** *Consider the reference model $\widehat{P}_{h,a,t} = \Phi \widehat{M}_{h,a,t}$ with $\widehat{M}_{h,a,t}$ defined in Equation (10). Then, for any $a \in \mathcal{A}, h \in [H]$, episode $t$ and any fixed function $g : \mathcal{S} \to [-H, H]$, the following holds with probability at least $1 - \delta$:*

$$\left\| \left( M_{h,a} - \widehat{M}_{h,a,t} \right) g \right\|_{\Sigma_{h,a,t-1}} \le 2H\sqrt{d \log (1 + tR^2/\lambda) + \log(1/\delta)} + C_P H \sqrt{\lambda d}.$$

*Proof.* We start by rewriting

$$\left\| \left( M_{h,a} - \widehat{M}_{h,a,t} \right) g \right\|_{\Sigma_{h,a,t-1}} = \left\| \Sigma_{h,a,t-1} \left( M_{h,a} - \widehat{M}_{h,a,t-1} \right) g \right\|_{\Sigma_{h,a,t-1}^{-1}},$$

and proceed by using the definitions of $\widehat{M}_{h,a,t}$, $\Sigma_{h,a,t-1}$ and $W_{h,a,t-1}$ to see that

$$\Sigma_{h,a,t-1}\left(M_{h,a} - \widehat{M}_{h,a,t}\right)g = \Phi^\mathsf{T} W_{h,a,t-1}\Phi M_{h,a}g + \lambda M_{h,a}g$$

$$- \Sigma_{h,a,t-1}\Sigma_{h,a,t-1}^{-1}\sum_{k=1}^{t-1}\mathbb{I}_{\{a_{h,k}=a\}}\varphi(x_{h,k})g\left(x_{h+1,k}\right)$$

$$= \Phi^\mathsf{T} W_{h,a,t-1}P_{h,a}g - \sum_{k=1}^{t-1}\mathbb{I}_{\{a_{h,k}=a\}}\varphi(x_{h,k})g\left(x_{h+1,k}\right) + \lambda M_{h,a}g$$

$$= \sum_{k=1}^{t-1}\mathbb{I}_{\{a_{h,k}=a\}}\left(\langle P_h(\cdot|x_{h,k},a_{h,k}),g\rangle - g(x_{h+1,k})\right)\varphi(x_{h,k}) + \lambda M_{h,a}g.$$

The first term on the right-hand side is a vector-valued martingale for an appropriately chosen filtration, since

$$\mathbb{E}\left[\langle P_h(\cdot|x_{h,k},a_{h,k}),g\rangle - g(x_{h+1,k})\,\middle|\, x_{h,k}, a_{h,k}\right] = 0,$$

so the sum of these terms can be bounded by appealing to Theorem 1 of Abbasi-Yadkori et al. [1] as

$$\left\|\sum_{k=1}^{t-1}\mathbb{I}_{\{a_{h,k}=a\}}\left(\langle P_h(\cdot|x_{h,k},a_{h,k}),g\rangle - g(x_{h+1,k})\right)\varphi^\mathsf{T}(x_{h,k})\right\|_{\Sigma_{h,a,t-1}^{-1}}$$

$$\leq 2H\sqrt{d\log\left(1 + tR^2/\lambda\right) + \log(1/\delta)}.$$

The proof is concluded by applying the bound

$$\|\lambda M_{h,a}g\|_{\Sigma_{h,a,t-1}^{-1}} \leq \sqrt{\lambda}\|M_{h,a}g\| \leq C_P H\sqrt{\lambda d},$$

where in the last step we used the assumption that $\|m_{h,a}(x)\|_1 \leq C_P$ and $\|g\|_\infty \leq H$. $\qquad\square$

The following simple result will also be useful in bounding the sum of exploration bonuses and thus the regret of the two algorithms:

**Lemma 16.** *For any $h \in [H]$,*

$$\sum_{a\in\mathcal{A}}\sum_{t=1}^{K}\left\|\mathbb{I}_{\{a_{h,t}=a\}}\varphi(x_{h,t})\right\|_{\Sigma_{h,a,t-1}^{-1}} \leq 2\sqrt{dAK\log\left(1 + KR^2/\lambda\right)}.$$

*Proof.* The claim is directly proved by the following simple calculations:

$$\sum_{a\in\mathcal{A}}\sum_{t=1}^{K}\left\|\mathbb{I}_{\{a_{h,t}=a\}}\varphi(x_{h,t})\right\|_{\Sigma_{h,a,t-1}^{-1}} \leq \sqrt{\sum_{a}\sum_{t=1}^{K}\mathbb{I}_{\{a_{h,t}=a\}}}\sqrt{\sum_{a}\sum_{t=1}^{K}\left\|\mathbb{I}_{\{a_{h,t}=a\}}\varphi(x_{h,t})\right\|_{\Sigma_{h,a,t}^{-1}}^2}$$

$$\leq 2\sqrt{K\sum_{a}\log\left(\frac{\det\left(\Sigma_{h,a,K}\right)}{\det\left(\lambda I\right)}\right)} \leq 2\sqrt{KdA\log\left(1 + KR^2/\lambda\right)},$$

where the first inequality is Cauchy–Schwarz and the second one follows from Lemma 11 of Abbasi-Yadkori et al. [1]. $\qquad\square$

Finally, the following result will be useful to bound the scale of the esimated model $\widehat{M}_{h,a,t}$ with probability 1:

**Lemma 17.** *Consider the reference model $\widehat{P}_{h,a,t} = \Phi\widehat{M}_{h,a,t}$ with $\widehat{M}_{h,a,t}$ defined in Equation* (10). *Then, for any $B > 0$ and any fixed function $g : \mathcal{S} \to [-B, B]$, the following statements hold with probability 1:*

$$\left\|\widehat{M}_{h,a,t}g\right\| \leq \frac{tBR}{\lambda} \qquad and \qquad \left\|\left(M_{h,a} - \widehat{M}_{h,a,t}\right)g\right\|_{\Sigma_{h,a,t-1}} \leq \lambda^{-1/2}tBR + \lambda^{1/2}BC_P.$$

*Proof.* The first statement is proven by straightforward calculations, using the definition of $\widehat{M}_{h,a,t}$:

$$\left\| \widehat{M}_{h,a,t} g \right\| = \left\| \Sigma_{h,a,t-1}^{-1} \sum_{k=1}^{t-1} \mathbb{I}_{\{a_{h,k}=a\}} \varphi(x_{h,k}) g\left(x_{h+1,k}\right) \right\|$$

$$\leq \left\| \Sigma_{h,a,t-1}^{-1} \right\|_{\text{op}} \left\| \sum_{k=1}^{t-1} \mathbb{I}_{\{a_{h,k}=a\}} \varphi(x_{h,k}) g\left(x_{h+1,k}\right) \right\| \leq \frac{B}{\lambda} \sum_{k=1}^{t-1} \|\varphi(x_{h,k})\| \leq \frac{tBR}{\lambda},$$

where the second inequality uses that the operator norm of $\Sigma_{h,a,t-1}^{-1}$ is at most $\lambda^{-1}$, and the triangle inequality. As for the second inequality, we proceed as in the proof of Proposition 15 and recall that

$$\Sigma_{h,a,t-1} \left( M_{h,a} - \widehat{M}_{h,a,t} \right) g = \sum_{k=1}^{t-1} \mathbb{I}_{\{a_{h,k}=a\}} \left( \langle P_h(\cdot|x_{h,k}, a_{h,k}), g \rangle - g(x_{h+1,k}) \right) \varphi(x_{h,k}) + \lambda M_{h,a} g.$$

The norm of the above is clearly bounded by $tBR + \lambda BC_P$. Thus, we have

$$\left\| \left( M_{h,a} - \widehat{M}_{h,a,t} \right) g \right\|_{\Sigma_{h,a,t-1}} = \left\| \Sigma_{h,a,t-1} \left( M_{h,a} - \widehat{M}_{h,a,t} g \right) \right\|_{\Sigma_{h,a,t-1}^{-1}}$$

$$\leq \left\| \Sigma_{h,a,t-1}^{-1/2} \right\|_{\text{op}} \left\| \Sigma_{h,a,t-1} \left( M_{h,a} - \widehat{M}_{h,a,t} \right) g \right\|$$

$$\leq \frac{1}{\sqrt{\lambda}} \left( tBR + \lambda BC_P \right) = \lambda^{-1/2} tBR + \lambda^{1/2} BC_P.$$

This concludes the proof. $\qquad\qquad\square$

## B.1 Optimism in state space through local confidence sets

This section presents our approach for factored linear MDPs with local confidence sets, which can be seen to lead to confidence bonuses in the state space. We first state some structural results that will justify our algorithmic approach, explain our algorithm in more detail, and then present the performance guarantees.

We recall that our approach is based on solving the following optimization problem:

$$\underset{q \in \mathcal{Q}(x_1), \omega, \widetilde{P}}{\text{maximize}} \quad \sum_{h=1}^{H} \sum_{a} \langle W_{h,a,t-1} \Phi \omega_{h,a}, r_a \rangle$$

$$\text{subject to} \quad \sum_{a} q_{h+1,a} = \sum_{a} \widetilde{P}_{h,a} W_{h,a,t-1} \Phi \omega_{h,a} \qquad \forall a \in \mathcal{A}, h = 1, \ldots, H$$

$$\Phi^{\mathsf{T}} q_{h,a} = \Phi^{\mathsf{T}} W_{h,a,t-1} \Phi \omega_{h,a} \qquad \forall a \in \mathcal{A}, h = 1, \ldots, H$$

$$D\left( \widetilde{P}_h(\cdot|x,a), \widehat{P}_{h,t}(\cdot|x,a) \right) \leq \epsilon_{h,t}(x,a) \qquad \forall(x,a),$$

where $D$ is an arbitrary divergence that is positive homogeneous and convex in its arguments. The following structural result shows that this optimization problem can be equivalently written in a dual form that is essentially identical to the optimistic Bellman equations derived in Section 4 for the tabular setting.

**Proposition 18.** *The optimization problem above is equivalent to solving the optimistic Bellman equations* (11) *with the exploration bonus defined as*

$$\text{CB}_h(x,a) = D^* \left( V_{h+1}^+ \middle| \epsilon_h(x,a), \widehat{P}_h(\cdot|x,a) \right).$$

The proof follows from a similar reparametrization as used in the proof of Proposition 1 that makes the optimization problem convex, thus enabling us to establish strong duality. To maintain readability, we defer the proof to Appendix B.3.1. Consequently, the properties stated in Propositions 2 and 3 can also be shown in a straightforward fashion.

Our results are based on using the divergence measure

$$D\left( \widetilde{P}_{h,t}(\cdot|x,a), \widehat{P}_{h,t}(\cdot|x,a) \right) = \sup_{g \in \mathcal{V}_{h+1,t}} \left\langle \widetilde{P}_{h,t}(\cdot|x,a) - \widehat{P}_{h,t}(\cdot|x,a), g \right\rangle,$$

whose conjugate can be directly upper-bounded by $\epsilon_{h,t}$. Since the structural results established above directly imply that Theorem 4 continues to hold, we can easily derive a practical and effective algorithm by simply using $\epsilon_{h,t}$ as the exploration bonuses. Specifically, we will consider an algorithm that calculates an optimistic value function and a corresponding policy by solving the OPB equations (11) via dynamic programming, with the confidence bonuses chosen as

$$\text{CB}_{h,t}^{\dagger}(x, a) = \alpha_{h,t} \, \|\varphi(x)\|_{\Sigma_{h,a,t-1}^{-1}}$$

for some $\alpha_{h,t}$. The shape of this confidence set is directly motivated by the following simple corollary of our general concentration result in Lemma 15:

**Lemma 19.** *Fix $h, a$ and consider the reference model $\widehat{P}_{h,a,t} = \Phi\widehat{M}_{h,a,t}$ with $\widehat{M}_{h,a,t}$ defined in Equation* (10)*. Then, for any fixed function $g : \mathcal{S} \to [-H, H]$, the following holds simultaneously for all $x$ under event $\mathcal{E}_{h,a,t}(g, \delta)$:*

$$\left\langle P_h(\cdot|x, a) - \widehat{P}_{h,t}(\cdot|x, a), g \right\rangle \leq C_t(\delta) \, \|\varphi(x)\|_{\Sigma_{h,a,t-1}^{-1}}.$$

*Proof.* The proof is immediate using the definition of the event $\mathcal{E}_{h,a,t}(g, \delta)$ and the Cauchy–Schwarz inequality:

$$\left\langle P_h(\cdot|x, a) - \widehat{P}_{h,t}(\cdot|x, a), g \right\rangle = \left\langle \varphi(x), \left( M_{h,a} - \widehat{M}_{h,a,t} \right) g \right\rangle$$

$$\leq \|\varphi(x)\|_{\Sigma_{h,a,t-1}^{-1}} \left\| \left( M_{h,a} - \widehat{M}_{h,a,t} \right) g \right\|_{\Sigma_{h,a,t-1}} \leq C_t(\delta) \, \|\varphi(x)\|_{\Sigma_{h,a,t-1}^{-1}}.$$

$\square$

The main challenge in the analysis will be to show that there exists an appropriate choice of $\alpha_{h,t}$ that guarantees that the above result holds uniformly over the value-function class $\mathcal{V}_{h+1,t}$ used in the definition of the confidence sets. We note that the resulting algorithm is essentially identical to the LSVI-UCB algorithm proposed and analyzed by Jin et al. [26], and we will accordingly refer to it by this name (that stands for "least-squares value iteration with upper confidence bounds").

### B.1.1 Regret Bound

In this section we prove the regret bound of Theorem 8, whose precise statement is as follows:

**Theorem 20.** *With probability greater than $1 - \delta$, the regret of LSVI-UCB with the choice $\lambda = 1$ and*

$$\alpha_{h,t} = \alpha = 2H \sqrt{d \log\left(1 + KR^2\right) + \log(HA/\delta) + dA\left(\log(1 + 4HK^2R^2) + d\log(1 + 4R^3K^3)\right)}$$

$$+ C_P \left( H\sqrt{d} + 1 \right) + 1$$

*can be bounded as*

$$\mathfrak{R}_T = \widetilde{O}(A\sqrt{H^3 d^3 T}).$$

We note that the statement of the theorem is trivial when $\alpha > K$ so we will suppose that the contrary holds throughout the analysis. The proof is a straightforward application of Theorem 4: given that $P \in \mathcal{P}$, the regret is bounded by the sum of exploration bonuses, which itself can be easily bounded using Lemma 16. Thus, the main challenge is to show that the transition model lies in the confidence set. To prove this, we observe that, thanks to the choice of exploration bonus, the class of value functions $\mathcal{V}_{h+1,t}$ produced by the algorithm is composed of functions of the form

$$V_{t,h}^{+}(x) = \min\left\{ H - h, \, \max\left\{ \langle\varphi(x), \theta_{t,a,h}\rangle + \alpha \, \|\varphi(x)\|_{\Sigma_{t,a,h}^{-1}} \right\} \right\},$$

and the covering number of this class is relatively small. We formalize this in the following proposition, which takes care of the probabilistic part of the analysis:

**Proposition 21.** *Consider the reference model $\widehat{P}_{h,a,t} = \Phi\widehat{M}_{h,a,t}$ with $\widehat{M}_{h,a,t}$ defined in Equation* (10)*. Then, for the choice of $\alpha$ in Theorem 20, the following holds simultaneously for all $x, a, h, t$, with probability at least $1 - \delta$:*

$$\sup_{V \in \mathcal{V}_{h+1,t}} \left\langle P_h(\cdot|x, a) - \widehat{P}_{h,t}(\cdot|x, a), V \right\rangle \leq \alpha \, \|\varphi(x)\|_{\Sigma_{h,a,t-1}^{-1}}.$$

The proof of this statement is rather technical and borrows some elements of the analysis of Jin et al. [26]—we delegate the proof to Appendix B.3.2. Thus, we now have all the necessary ingredients to conclude the proof of Theorem 20. Indeed, since Proposition 21 guarantees that the true model $P$ is always in the confidence set with probability $1 - \delta$, and using the optimistic property of our algorithm that follows from Proposition 18, we can appeal to Theorem 5 to bound the regret in terms of the sum of exploration bonuses. This in turn can be bounded by using Lemma 16 as follows:

$$\sum_{h=1}^{H} \sum_{t=1}^{K} \mathrm{CB}_{h,t}^{\dagger}(x_{h,t}, a_{h,t}) \leq \sum_{h=1}^{H} \sum_{a} \sum_{t=1}^{K} \left\| \mathbb{I}_{\{a_{h,t}=a\}} \varphi(x_{h,t}) \right\|_{\Sigma_{h,a,t-1}^{-1}} \alpha_{h,a,t}$$

$$\leq 2\alpha H \sqrt{dAK \log\left(1 + KR^2/\lambda\right)} = 2\alpha \sqrt{HdAT \log\left(1 + KR^2/\lambda\right)}.$$

The proof is concluded by observing that $\alpha = \widetilde{O}(Hd\sqrt{A})$.

## B.2    Optimism in feature space through global constraints

We now present our approach based on global confidence sets for the transition model $\widetilde{M}$ that lead to an algorithm using exploration bonuses that can be expressed in the feature space. The main idea behind the algorithm is defining in each episode $t$, the confidence set $\mathcal{M}_t$ of models $\widetilde{M}$ satisfying

$$D(\widetilde{M}_{h,a}, \widehat{M}_{h,a,t}) = \sup_{f \in \mathcal{V}_{h+1}} \left\| (\widetilde{M}_{h,a} - \widehat{M}_{h,a,t}) f \right\|_{\Sigma_{h,a,t-1}} \leq \epsilon_{h,a,t}$$

for an appropriate choice of $\epsilon_{h,a,t}$, and defining the function

$$G_t(\widetilde{M}) = \max_{q \in \mathcal{Q}(x_1), \omega} \sum_{h=1}^{H} \sum_{a} \langle W_{h,a,t-1} \Phi \omega_{h,a}, r_a \rangle \tag{31}$$

$$\text{subject to} \quad \sum_{a} q_{h-1,a} = \sum_{a} \widetilde{M}_{h,a}^{\mathsf{T}} \Phi^{\mathsf{T}} W_{h,a,t-1} \Phi \omega_{h,a} \qquad \forall a \in \mathcal{A}, h = 1, \dots, H$$

$$\Phi^{\mathsf{T}} q_{h,a} = \Phi^{\mathsf{T}} W_{h,a,t-1} \Phi \omega_{h,a} \qquad \forall a \in \mathcal{A}, h = 1, \dots, H.$$

Clearly, if the true model $M$ is in the confidence set $\mathcal{M}_t$, we have $\max_{\widetilde{M} \in \mathcal{M}_t} G_t(\widetilde{M}) \geq G_t(M) = V_1^*(x_1)$. As phrased above, this optimization problem is intractable due to the large number of variables and constraints. Our algorithm addresses this challenge by converting the above problem into a more tractable one that retains the optimistic property. In particular, our algorithm solves the parametric OPB equations (11) with confidence bonuses defined as

$$\mathrm{CB}_{h,t}^{\dagger}(x, a) = \langle \varphi(x), B_{h,a,t}^{\dagger} \rangle$$

for a vector $B_{h,a,t}^{\dagger} \in \mathbb{R}^d$ chosen to maximize the following function over the convex set $\mathcal{B}_t = \{B : \|B_{h,a}\|_{\Sigma_{h,a,t-1}} \leq \epsilon_{h,a,t}\}$:

$$G_t'(B) = \max_{q \in \mathcal{Q}(x_1), \omega} \sum_{h=1}^{H} \sum_{a} \langle W_{h,a,t-1} \Phi \omega_{h,a}, r_a + \Phi B_{h,a} \rangle \tag{32}$$

$$\text{subject to} \quad \sum_{a} q_{h-1,a} = \sum_{a} \widehat{M}_{h,a,t}^{\mathsf{T}} \Phi^{\mathsf{T}} W_{h,a,t-1} \Phi \omega_{h,a} \qquad \forall a \in \mathcal{A}, h = 1, \dots, H$$

$$\Phi^{\mathsf{T}} q_{h,a} = \Phi^{\mathsf{T}} W_{h,a,t-1} \Phi \omega_{h,a} \qquad \forall a \in \mathcal{A}, h = 1, \dots, H.$$

This definition is easily seen to be equivalent to the one given in the statement of Theorem 9 through basic LP duality (cf. Section 2). Our analysis will take advantage of the fact that our exploration bonuses are linear in the feature representation, which eventually yields value functions of the following form:

$$V_{h,t}^{\dagger}(x) = \min \left\{ H - h + 1, \max_a \langle \varphi(x), \theta_{h,a,t}^{\dagger} \rangle \right\}, \tag{33}$$

for some $\theta_{h,a,t}^{\dagger} \in \mathbb{R}^d$, which implies that the class of functions $\mathcal{V}_{h+1,t}$ is simpler than in the case LSVI-UCB. The algorithm is justified by the following property:

**Proposition 22.** *For any episode $t$, let the functions $G_t$ and $G'_t$ be defined as above and let $\mathcal{B}_t = \left\{ B : \|B_{h,a}\|_{\Sigma_{h,a,t-1}} \leq \epsilon_{h,a,t} \right\}$. Then, $\max_{B \in \mathcal{B}_t} G'_t(B) \geq \max_{\widetilde{M} \in \mathcal{M}_t} G_t(\widetilde{M})$.*

*Proof.* Let us fix a model $\widetilde{M} \in \mathcal{M}_t$, introduce the notation $Z_{h,a,t} = \left( \widetilde{M}_{h,a} - \widehat{M}_{h,a,t} \right) V_{h+1,t}$, and notice that $Z_t \in \mathcal{B}_t$ due to the definition of $\mathcal{M}_t$. The proof relies on expressing the values of $G_t(\widetilde{M})$ and $G'_t(B)$ through the OPB equations (11) defining them. Indeed, for a fixed $\widetilde{M}$, the value of $G_t(\widetilde{M})$ can be expressed through standard LP duality as exposed in Section 2. To express $G_t(\widetilde{M})$, let $U_t$ stand for the value function defined through the system of equations

$$
\begin{aligned}
\theta_{h,a,t} &= \rho_a + \widetilde{M}_{h,a} U_{h+1,t} = \rho_a + \left( \widetilde{M}_{h,a} - \widehat{M}_{h,a,t} \right) U_{h+1,t} + \widehat{M}_{h,a,t} U_{h+1,t} \\
&= \rho_a + Z_{h,a,t} + \widehat{M}_{h,a,t} U_{h+1,t}, \\
U_{h+1,t}(x) &= \max_a \langle \varphi(x), \theta_{h+1,a,t} \rangle
\end{aligned}
$$

that have to be satisfied for all $x, a, h$. Then, it is easy to see that $G_t(\widetilde{M}) = U_{1,t}(x_1)$. Notice that this can be understood as the solution of the OPB equations (11) with exploration bonus $\mathrm{CB}_{h,t}(x,a) = \langle \varphi(x), Z_{h,a,t} \rangle$. On the other hand, $G'_t(B)$ can be expressed as $U'_{1,t}(x_1)$ with $U'_t$ is defined through the system of equations

$$
\begin{aligned}
\theta'_{h,a,t} &= \rho_a + \widetilde{M}_{h,a,t} U'_{h+1,t} \\
U'_{h,t}(x) &= \max_a \langle \varphi(x), \theta'_{h,a,t} + B_{h,a,t} \rangle .
\end{aligned}
$$

It is then easy to verify that $G_t(\widetilde{M}) = G'_t(Z)$ and, using $Z \in \mathcal{B}_t$, that $G'_t(Z) \leq \max_{B \in \mathcal{B}_t} G'_t(B)$. This concludes the proof since the inequality must hold for any model $\widetilde{M} \in \mathcal{M}_t$. $\qquad\square$

Notably, the above proposition ensures that the value function $V_t^\dagger$ arising from the OPB equations (11) with bonus $\mathrm{CB}_{h,t}^\dagger(x,a) = \langle \varphi(x), B_{h,a,t}^\dagger \rangle$ is optimistic in the sense that $V_{1,t}^\dagger(x_{1,t}) \geq G_t(\widetilde{M}) \geq V_1^*(x_{1,t})$. This enables us to apply the general regret bound of Theorem 5 to establish a performance guarantee for the resulting algorithm. We provide this analysis in the next section.

From the above formulation, it is readily apparent that, since $G'$ is a maximum of linear functions, it is a convex function of $B$, and thus maximizing it over a convex set is potentially still very challenging. We note that this optimization problem is essentially identical to the one faced by the seminal LinUCB algorithm for linear bandits [15, 1], which is known to be computationally intractable for general decision sets. This is to be contrasted with the algorithms described in previous parts of this paper, which are efficiently implementable through dynamic programming. Indeed, despite being of a similar form, the simplicity of these previous methods stem from the local nature of their confidence sets which was seen to lead to exploration bonuses that can be set independently for each state and computed via dynamic programming. This is no longer possible for the exploration bonuses used in this section, which are set through a global parameter vector $B$. Intuitively, this prevents the application of dynamic-programming methodology which heavily relies on the ability of breaking down an optimization problem into a set of local optimization problems (often referred to as the "principle of optimality" in this context [9]). It remains an open problem to find an efficient implementation of this method.

It is interesting to note that our algorithm essentially coincides with the ELEANOR method proposed very recently by Zanette et al. [54], up to minor differences. Their analysis is more general than ours as they considered the significantly harder case of learning with misspecified linear models that our analysis doesn't account for. Nevertheless, our analysis is substantially simplified by our model-based perspective that sheds new light on the algorithm. In particular, while Zanette et al. [54] do not provide a substantial discussion of the computational challenges associated with ELEANOR, our formulation clearly highlights the convexity of the objective function optimized by the algorithm and the relation with LinUCB. We believe that our model-based perspective can provide further insights into this challenging problem in the future, and particularly that it will remain useful when analyzing misspecified linear models.

### B.2.1 Regret bound

We now prove our main result regarding the algorithm: the regret bound claimed in Theorem 9. In particular, the detailed statement of this result is as follows:

**Theorem 23.** *With probability greater than $1 - \delta$, the regret of our algorithm with $\lambda = 1$ for*

$$\epsilon_{h,a,t} = \epsilon = 2H\sqrt{d\log\left(1 + KR^2\right) + dA\log(1 + 4K^2HR^3) + \log\left(HA/\delta\right)}$$
$$+ \lambda^{1/2}\left(C_P\sqrt{d} + 1 + C_P\right)$$

*satisfies*

$$\mathfrak{R}_T = \widetilde{O}(dA\sqrt{H^3T}).$$

The key idea of the analysis is to use Proposition 22 to establish the optimistic property of the algorithm and use Theorem 5 to bound the regret by the sum of exploration bonuses. The only remaining challenge is to prove that, with high probability, the true model lies in the confidence sets specified in Equation (14). The following proposition guarantees that this is indeed true:

**Proposition 24.** *Consider the reference model $\widehat{P}_{h,a,t} = \Phi\widehat{M}_{h,a,t}$ with $\widehat{M}_{h,a,t}$ defined in Equation (10). Then, for the choice of $\epsilon$ in Theorem 23, the following holds simultaneously for all $a, h, t$, with probability at least $1 - \delta$:*

$$\sup_{f\in\mathcal{V}_{h+1,t}} \left\|\left(M_{h,a} - \widehat{M}_{h,a,t}\right)f\right\|_{\Sigma_{h,a,t-1}} \le \epsilon_{h,a,t}.$$

The proof relies on a covering argument similar to the one we used for proving Proposition 21, exploiting the fact that the value function class $\mathcal{V}_{h+1,t}$ is composed of slightly simpler functions. The proof is deferred to Appendix B.3.4. Thus, we can conclude the proof of Theorem 23 as follows. Taking advantage of the fact that the algorithm follows the optimal policy corresponding to the solution of the OPB equations (11), we can use the general guarantee of Theorem 5 and bound the regret of the algorithm as the sum of the exploration bonuses. Noticing that the bonuses can be upper-bounded as

$$\text{CB}^{\dagger}_{h,t}(x,a) = \left\langle\varphi(x), B^{\dagger}_{h,a,t}\right\rangle \le \|\varphi(x_{h,t})\|_{\Sigma^{-1}_{h,a,t-1}} \left\|B^{\dagger}_{h,a,t}\right\|_{\Sigma_{h,a,t-1}} \le \|\varphi(x_{h,t})\|_{\Sigma^{-1}_{h,a,t-1}} \epsilon_{h,a,t},$$

where the last step follows from the fact that $B^{\dagger}_{h,a,t} \in \mathcal{B}_t$, the sum of confidence bonuses can be bounded by appealing to Lemma 16:

$$\sum_{h=1}^{H}\sum_{t=1}^{K}\text{CB}^{\dagger}_{h,t}(x_{h,t}, a_{h,t}) \le \sum_{h=1}^{H}\sum_{a}\sum_{t=1}^{K}\left\|\mathbb{I}_{\{a_{h,t}=a\}}\varphi(x_{h,t})\right\|_{\Sigma^{-1}_{h,a,t-1}} \epsilon_{h,a,t}$$

$$\le 2\epsilon H\sqrt{dAK\log\left(\frac{1 + KR^2/\lambda}{\delta}\right)} = 2\epsilon\sqrt{HdAT\log\left(\frac{1 + KR^2/\lambda}{\delta}\right)}.$$

Setting $\lambda = 1$ and noticing that $\epsilon = \widetilde{O}(H\sqrt{dA})$ concludes the proof of Theorem 23.

### B.3 Technical proofs

### B.3.1 Proof of Proposition 18

We first note that, since $\widehat{P}_{h,a} = \Phi\widehat{M}_{h,a}$ and using the second constraint, the first constraint in the optimization problem can be rewritten as

$$\sum_{a} q_{h+1,a} = \sum_{a}\widehat{M}_{h,a}\Phi W_{h,a}\Phi\omega_{h,a} + \sum_{a}\left(\widetilde{P}_{h,a} - \widehat{P}_{h,a}\right)q_{h,a}.$$

Using this, we use a similar argument to Lemma 10 to show that strong duality and the KKT conditions hold. We reparameterize by defining $J_h(x, a, x') = q_h(x, a)\widetilde{P}_h(x'|x, a)$ and observe that the last constraint in (12) is can be written as $D(J_h(x, a, \cdot), \widehat{P}_h(\cdot|x, a)\sum_{x'} J_h(x, a, x')) \le \epsilon_h(x, a)\sum_{x'} J_h(x, a, x')$ which is convex in $J$. It can also be easily observed that the first two

constraints, and the objective are linear in $q, J, \omega$. Thus strong duality holds, and the optimal value of the reparameterized optimization problem is equal to the optimal value of the corresponding Lagrangian dual problem. As in the proof of Lemma 10, by using the reverse reparameterization, we can see that the value of the Lagrangian of the modified problem is equal to that of the original problem in (12). Hence, strong duality holds for (12). It then follows that the KKT conditions also hold for this problem.

Given strong duality, we can find the dual of the problem in (12) by considering the Lagrangian. The partial Lagrangian of the optimization problem without the last constraint of the primal can be written as

$$\mathcal{L}(q, \kappa, \omega; V, \theta) = \sum_{h,a} \left\langle W_{h,a} \Phi \omega_{h,a}, r_a + \widehat{P}_{h,a} V_{h+1} - \Phi \theta_{h,a} \right\rangle$$
$$+ \sum_{x,a,h} q_h(x,a) \left( (\Phi \theta_{h,a})(x) + \sum_y \kappa_h(x,a,y) V_{h+1}(y) - V_h(x) \right) + V_1(x_1),$$
(34)

for $\kappa_h(x,a,y) = \widetilde{P}_h(y|x,a) - \widehat{P}_h(y|x,a)$. Then, by strong duality, the optimal value of the primal is equal to

$$\min_{V,\theta} \max_{\substack{q \geq 0, \widetilde{P} \in \mathcal{P} \\ \omega, \kappa}} \mathcal{L}(q, \kappa, \omega; V, \theta).$$

Observing that $q_h(x,a) \geq 0$ and using the definition of $\kappa_h(x,a,\cdot)$, we can consider the inner maximization over $\widetilde{P}_h(\cdot|x,a) \in \mathcal{P}_h(x,a)$. We get,

$$\max_{\widetilde{P}_h(\cdot|x,a) \in \mathcal{P}_h(x,a)} \sum_y (\widetilde{P}_h(y|x,a) - \widehat{P}_h(y|x,a)) V_{h+1}(y) = D_*(V_{h+1}|\widehat{P}, \epsilon)$$

by definition of the conjugate. Substituting this back into (34), we can find the dual from this Lagrangian by a similar technique to Proposition 1. In particular, observe that the objective function will be given by $V_1(x_1)$. To define the constraints, note that if $\max_\omega \sum_{h,a} \left\langle W_{h,a} \Phi \omega_{h,a}, r_a + \widehat{P}_{h,a} V_{h+1} - \Phi \theta_{h,a} \right\rangle < \infty$, it must be the case that $\left\langle W_{h,a} \Phi, r_a + \widehat{P}_{h,a} V_{h+1} - \Phi \theta_{h,a} \right\rangle = 0$, and likewise if $\max_{q>0} \sum_{x,a,h} q_h(x,a)((\Phi \theta_{h,a})(x) + D_*(V_{h+1}|\epsilon_{h,a}, \widehat{P}_h(\cdot|x,a)) - V_h(x)) < \infty$, it must be the case that $(\Phi \theta_{h,a})(x) + D_*(V_{h+1}|\epsilon_{h,a}, \widehat{P}_h(\cdot|x,a)) - V_h(x) \leq 0$.

Thus the dual optimization problem can be written,

$$\underset{V}{\text{minimize}} \quad V_1(x_1)$$

$$\text{Subject to} \quad V_h(x) \geq (\Phi \theta_{h,a})(x) + D_* \left( V_{h+1} \middle| \epsilon_h(x,a), \widehat{P}_h(\cdot|x,a) \right) \qquad \forall (x,a) \in \mathcal{Z}, h \in [H]$$

$$(\Phi W_{h,a} \Phi) \theta_{h,a} = \Phi^\mathsf{T} W_{h,a} \left( r_a + \widehat{P}_{h,a} V_{h+1} \right) \qquad \forall a \in \mathcal{A}, h \in [H].$$

It is easily seen that the solution to this can be found by solving the optimistic parametric Bellman equations in (11) with $\text{CB}_{h,t}(x,a) = D_* \left( V_{h+1,t} \middle| \epsilon_{h,t}(x,a), \widehat{P}_{h,t}(\cdot|x,a) \right)$ via backwards recursion.  $\square$

### B.3.2  The proof of Proposition 21

The proof follows from a construction proposed by Jin et al. [26]: it relies on taking a union bound over an appropriately chosen covering of the class of value functions in stage $h+1$ that can be ever produced by solving the optimistic Bellman equations (11). For this purpose, we need the following technical result that bounds the covering number of this set:

**Lemma 25.** *Let $\mathcal{N}(\mathcal{V}, \varepsilon)$ be the $\varepsilon$-covering number of the set $\mathcal{V}$ with respect to the distance $\|V - V'\|_\infty = \sup_{x \in \mathcal{S}} |V(x) - V'(x)|$. Then, for any stage $h = 1, \ldots, H$ and episode $t$,*

$$\log(\mathcal{N}(\mathcal{V}_{h+1,t}, \varepsilon)) \leq Ad \log(1 + 4tHR/(\lambda \varepsilon)) + d^2 A \log(1 + 4R\alpha/(\lambda \varepsilon^2))$$

*where $R$ is such that $\|\varphi(x)\|_2 \le R \,\forall x \in \mathcal{S}$, $\lambda$ is such that the minimum eigenvalue, $\lambda_{\min}(\Sigma_{h,a,t}) \ge \lambda \,\forall a \in \mathcal{A}, h \in [H], t \in [K]$.*

The proof of Lemma 25 is similar to that of Lemma D.6 of [26], and exploits that the class $\mathcal{V}_{h+1,t}$ is parametrized smoothly by $\theta$ and $\Sigma$. We relegate the proof to Appendix B.3.3. As for the proof of Proposition 21, let us fix any $h, a, \varepsilon > 0$ and any $V \in \mathcal{V}_{h+1,t}$, and let $\widetilde{V}$ be in the $\varepsilon$-covering of $\mathcal{V}_{h+1,t}$ defined in Lemma 25 such that $\|V - \widetilde{V}\|_\infty \le \varepsilon$. Then, we have

$$
\left\langle P_h(\cdot|x,a) - \widehat{P}_{h,t}(\cdot|x,a), V \right\rangle
$$
$$
= \left\langle P_h(\cdot|x,a) - \widehat{P}_{h,t}(\cdot|x,a), \widetilde{V} \right\rangle + \left\langle P_h(\cdot|x,a) - \widehat{P}_{h,t}(\cdot|x,a), V - \widetilde{V} \right\rangle.
$$

The second term can be bounded by introducing the notation $\widetilde{g} = V - \widetilde{V}$ and writing

$$
\left\langle P_h(\cdot|x,a) - \widehat{P}_{h,t}(\cdot|x,a), \widetilde{g} \right\rangle = \left\langle \varphi(x), \left(M_h - \widehat{M}_{h,a,t}\right)\widetilde{g} \right\rangle \le \|\varphi(x)\|_{\Sigma_{h,a,t}^{-1}} \left\|\left(M_h - \widehat{M}_{h,a,t}\right)\widetilde{g}\right\|_{\Sigma_{h,a,t}}
$$
$$
\le \varepsilon \left(\lambda^{-1/2} tR + \lambda^{1/2} C_P\right)\|\varphi(x)\|_{\Sigma_{h,a,t}^{-1}},
$$

where we used Lemma 17 with $B = \varepsilon$ in the last step. As for the first term, we use a union bound over all $\widetilde{V}$ in the $\varepsilon$-covering of $\mathcal{V}_{h+1,t}$ and Lemma 19. Denoting the covering number as $\mathcal{N}_\varepsilon$ and setting $\delta' = \delta/HA$, we can see that for any $\widetilde{V}$ in the $\varepsilon$-covering, with probability greater than $1 - \delta'$, we have

$$
\left\langle P_h(\cdot|x,a) - \widehat{P}_{h,t}(\cdot|x,a), \widetilde{V} \right\rangle \le \|\varphi(x)\|_{\Sigma_{h,a,t}^{-1}} C_t(\delta'/\mathcal{N}_\varepsilon),
$$

which can be further bounded as

$$
C_t(\delta'/\mathcal{N}_\varepsilon) - C_P H\sqrt{\lambda d} = 2H\sqrt{d\log\left(1 + tR^2/\lambda\right) + \log\left(\mathcal{N}_\varepsilon/\delta'\right)}
$$
$$
\le 2H\sqrt{d\log\left(1 + tR^2/\lambda\right) + \log(1/\delta') + dA\log(1 + 4HtR/(\lambda\varepsilon)) + d^2 A\log(1 + 4R\alpha/(\lambda\varepsilon^2))}
$$
$$
\le 2H\sqrt{d\log\left(1 + tR^2/\lambda\right) + \log(1/\delta') + dA\log(1 + 4Ht^2R^2/\lambda^2) + d^2 A\log(1 + 4R^3Kt^2/\lambda^3)},
$$

where we set $\varepsilon = \lambda/(tR)$ and used the condition $\alpha \le K$ in the last step. With the same choice of $\varepsilon$, we also have

$$
\varepsilon\left(\lambda^{-1/2} tR + \lambda^{1/2} C_P\right) \le \lambda^{1/2}\left(1 + C_P\right).
$$

Noticing that the sum of the two latter terms is bounded by $\alpha$ and taking a union bound over all $h, a$ concludes the proof. $\qquad\square$

### B.3.3 The proof of Lemma 25

We first note that, due to the definition of the parameter vectors $\theta_{h,a,t}^+$ as the solution of the OPB equations (11) with $\left\|V_{h+1,t}^+\right\|_\infty \le H$, we have

$$
\left\|\theta_{h,a,t}^+\right\| \le \frac{tHR}{\lambda} \overset{\text{def}}{=} \beta,
$$

where the inequality follows from Lemma 17. To preserve clarity of writing, we omit explicit references to $t$ below. By design of the algorithm, we can see that the value functions can be written with the help of the function $U_{h,\theta,\Sigma}$ defined as

$$
U_{h,\theta,\Sigma}(x) = \min\left\{H - h + 1, \max_{a\in\mathcal{A}}\left\{\langle\varphi(x), \theta_{h,a}\rangle + \alpha\|\varphi(x)\|_{\Sigma_{h,a}^{-1}}\right\}\right\}
$$

for some $\alpha > 0$. Indeed, the class of value functions can be written as

$$
\mathcal{V}_h = \left\{U_{h,\theta,\Sigma}: \ \max_a\|\theta_{h,a}\| \le \beta, \ \max_a\left\|\Sigma_{h,a}^{-1}\right\|_{\text{op}} \le 1/\lambda\right\}.
$$

We show below that $U_{h,\theta,\Sigma}$ is a smooth function of the parameters $\theta_{h,a}$ and $\Sigma_{h,a}^{-1}$, which will allow us to prove a tight bound on the covering number of the class $\mathcal{V}_h$. Indeed, letting $V_h = U_{h,\theta,\Sigma}$ and

$\widetilde{V}_h = U_{h,\widetilde{\theta},\widetilde{\Sigma}}$ for an arbitrary set of parameters $\theta, \Sigma, \widetilde{\theta}, \widetilde{\Sigma}$, we have

$$\|V_h - \widetilde{V}_h\|_\infty = \sup_{x \in \mathcal{S}} \left| \min\{H - h + 1, \max_{a \in \mathcal{A}}\{\varphi(x)^{\mathsf{T}}\theta_{h,a} + \alpha\|\varphi(x)\|_{\Sigma_{h,a}^{-1}}\}\} \right.$$

$$\left. - \min\{H - h + 1, \max_{a \in \mathcal{A}}\{\varphi(x)^{\mathsf{T}}\widetilde{\theta}_{h,a} + \alpha\|\varphi(x)\|_{\widetilde{\Sigma}_{h,a}^{-1}}\}\} \right|$$

$$\leq \sup_{x \in \mathcal{S}} \left| \max_{a \in \mathcal{A}}\{\varphi(x)^{\mathsf{T}}\theta_{h,a} + \alpha\|\varphi(x)\|_{\Sigma_{h,a}^{-1}}\} - \max_{a \in \mathcal{A}}\{\varphi(x)^{\mathsf{T}}\widetilde{\theta}_{h,a} + \alpha\|\varphi(x)\|_{\widetilde{\Sigma}_{h,a}^{-1}}\} \right|$$

$$\leq \sup_{x \in \mathcal{S}, a \in \mathcal{A}} \left| \varphi(x)^{\mathsf{T}}\theta_{h,a} + \alpha\|\varphi(x)\|_{\Sigma_{h,a}^{-1}} - \varphi(x)^{\mathsf{T}}\widetilde{\theta}_{h,a} + \alpha\|\varphi(x)\|_{\widetilde{\Sigma}_{h,a}^{-1}} \right|$$

$$\leq \sup_{x \in \mathcal{S}, a \in \mathcal{A}} \left| \varphi(x)^{\mathsf{T}}(\theta_{h,a} - \widetilde{\theta}_{h,a}) + \sqrt{\varphi(x)^{\mathsf{T}}(\alpha\Sigma_{h,a}^{-1} - \alpha\widetilde{\Sigma}_{h,a}^{-1})\varphi(x)} \right|$$

$$\leq \sup_{a \in \mathcal{A}} R\|\theta_{h,a} - \widetilde{\theta}_{h,a}\|_2 + \sup_{a \in \mathcal{A}} R\|\alpha\Sigma_{h,a}^{-1} - \alpha\widetilde{\Sigma}_{h,a}^{-1}\|_{\mathrm{op}}$$

$$\leq \sup_{a \in \mathcal{A}} R\|\theta_{h,a} - \widetilde{\theta}_{h,a}\|_2 + \sup_{a \in \mathcal{A}} R\alpha\|\Sigma_{h,a}^{-1} - \widetilde{\Sigma}_{h,a}^{-1}\|_F$$

since $\|\varphi(x)\|_2 \leq R$ and we have used $\|A\|_{\mathrm{op}}$ to denote the operator norm and $\|A\|_F$ the Frobenius norm of a matrix $A$.

We then note that the $\varepsilon/2$-covering number of the set $\Theta = \{(\theta_a)_{a \in \mathcal{A}} : \theta_a \in \mathbb{R}^d, \sup_{a \in \mathcal{A}} \|\theta_a\|_2 \leq \beta\}$ is bounded by $(1 + 4\beta/\varepsilon)^{Ad}$, and that $\varepsilon/2$-covering number of the set $\Gamma = \{(\Sigma_a)_{a \in \mathcal{A}} : \Sigma_a \in \mathbb{R}^{d \times d}, \sup_{a \in \mathcal{A}} \|\Sigma_a\|_F \leq 1/\lambda\}$ is bounded by $(1 + 4/(\lambda\varepsilon^2))^{d^2 A}$. These results follow due to the standard fact that the $\varepsilon$-covering number of a ball in $\mathbb{R}^d$ with radius $R > 0$ with $\ell_2$ distance is bounded by $(1 + 2R/\varepsilon)^d$, and that $\Theta$ and $\Gamma$ are $(dA)$-dimensional and $(d^2 A)$-dimensional, respectively.

From the above discussion, we can conclude that for any $V_h \in \mathcal{V}_h$, there is a $\widetilde{V}_h$ parameterized by $\widetilde{\theta}_h$ in the $\varepsilon/2$-covering of $\Theta_h$, and $\widetilde{\Sigma}_h$ in the $\varepsilon/2$-covering of $\Gamma_h$ such that,

$$\|V_h - \widetilde{V}_h\|_\infty \leq R\varepsilon/2 + R\alpha\varepsilon/2.$$

By rescaling of the covering numbers, we can see that the logarithm of the $\varepsilon$-covering number of $\mathcal{V}_h$ can be bounded by

$$\log(\mathcal{N}(\mathcal{V}_h, \varepsilon)) \leq \log(\mathcal{N}(\Theta_h, \varepsilon/(2R))) + \log(\mathcal{N}(\Gamma_h, \varepsilon/(2\alpha R)))$$

$$\leq Ad \log(1 + 4\beta R/\varepsilon) + d^2 A \log(1 + 4R\alpha/(\lambda\varepsilon^2)).$$

Substituting in $\beta = \frac{tHR}{\lambda}$ gives the result.

$\square$

### B.3.4   The proof of Proposition 24

The proof is similar to that of Proposition 21, in that it also relies on a covering argument to prove uniform convergence over the set of potential value functions. The following technical result bounds the covering number of this set:

**Lemma 26.** *Let $\mathcal{N}(\mathcal{V}, \varepsilon)$ be the $\varepsilon$-covering number of some set $\mathcal{V}$ with respect to the distance $\|V - V'\|_\infty = \sup_{x \in \mathcal{S}} |V(x) - V'(x)|$. Then, for any stage $h = 1, \ldots, H$ and episode $t$,*

$$\log(\mathcal{N}(\mathcal{V}_{h+1,t}, \varepsilon)) \leq dA \log(1 + 4tHR^2/(\varepsilon\lambda)).$$

To reduce clutter, we defer the proof to Appendix B.3.5. To proceed, we fix $h, a, \varepsilon > 0$ and an arbitrary $V \in \mathcal{V}_{h+1,t}$, and consider a $\widetilde{V}$ in the covering defined above such that $\left\|V - \widetilde{V}\right\|_\infty \leq \varepsilon$. Then, by the triangle inequality, we have

$$\left\|(M_{h,a} - \widehat{M}_{h,a,t})V\right\|_{\Sigma_{h,a,t-1}} \leq \left\|(M_{h,a} - \widehat{M}_{h,a,t})\widetilde{V}\right\|_{\Sigma_{h,a,t-1}} + \left\|(M_{h,a} - \widehat{M}_{h,a,t})(V - \widetilde{V})\right\|_{\Sigma_{h,a,t-1}}$$

$$\leq \left\|(M_{h,a} - \widehat{M}_{h,a,t})\widetilde{V}\right\|_{\Sigma_{h,a,t-1}} + \varepsilon\left(\lambda^{-1/2}tR + \lambda^{1/2}C_P\right),$$

where we used Lemma 17 with $B = \varepsilon$ in the last step. Setting $\delta' = \delta/(HA)$, the first term can be bounded with probability at least $1 - \delta'$ by exploiting that $\widetilde{V}$ is in the covering, and using the union bound to show that for every such $\widetilde{V}$, we simultaneously have

$$
\begin{aligned}
\left\|\left(M_{h,a} - \widehat{M}_{h,a,t}\right)\widetilde{V}\right\|_{\Sigma_{h,a,t-1}} &\leq C_t(\delta'/\mathcal{N}_\varepsilon) = 2H\sqrt{d\log\left(1 + tR^2/\lambda\right) + \log\left(\mathcal{N}_\varepsilon/\delta'\right)} + C_P H\sqrt{\lambda d} \\
&\leq 2H\sqrt{d\log\left(1 + tR^2/\lambda\right) + dA\log(1 + 4tHR^2/(\varepsilon\lambda)) + \log\left(1/\delta'\right)} + C_P H\sqrt{\lambda d}
\end{aligned}
$$

Putting the two bounds together and setting $\varepsilon = \lambda/(tR)$ gives

$$
\begin{aligned}
\left\|\left(M_{h,a} - \widehat{M}_{h,a,t}\right)V\right\|_{\Sigma_{h,a,t-1}} \leq{}& 2H\sqrt{d\log\left(1 + tR^2/\lambda\right) + dA\log(1 + 4t^2HR^3/\lambda^2) + \log\left(HA/\delta\right)} \\
&+ \lambda^{1/2}\left(C_P H\sqrt{d} + 1 + C_P\right).
\end{aligned}
$$

This is clearly upper-bounded by the chosen value of $\epsilon$. Taking a union bound over all $h, a$ concludes the proof. $\qquad\square$

### B.3.5   The proof of Lemma 26

The proof is similar to that of Lemma 25, although simpler due to the simpler form of the value functions in this case. We stary by noting that, due to the definition of the parameter vectors $\theta_{h,a,t}^+$ as the solution of the OPB equations (11) with $\left\|V_{h+1,t}^+\right\|_\infty \leq H$, we have

$$
\left\|\theta_{h,a,t}^+\right\| \leq \frac{tHR}{\lambda} \overset{\text{def}}{=} \beta,
$$

where the inequality follows from Lemma 17. Given the definition of the algorithm, it is easy to see that the value functions can be with the help of the function $U_{h,\theta}$ defined as

$$
U_{h,\theta}(x) = \min\left\{H - h + 1, \max_{a\in\mathcal{A}} \langle\varphi(x), \theta_{h,a}\rangle\right\},
$$

in the form $V_{h,t} = U_{h,\theta}$ for some $\theta$ with norm bounded by $\beta$. Thus, the set of value functions can be written as

$$
\mathcal{V}_h = \left\{U_{h,\theta}: \ \|\theta\| \leq \beta\right\}.
$$

We show below that $U$ is a smooth function of $\theta$, which will allow us to prove a tight bound on the covering number of the class $\mathcal{V}_h$. Indeed, this can be seen by

$$
\begin{aligned}
\left\|U_{h,\theta} - U_{h,\theta'}\right\|_\infty &\leq \sup_{x\in\mathcal{S}}\left|\max_{a\in\mathcal{A}} \langle\varphi(x), \theta_{h,a}\rangle - \max_{a\in\mathcal{A}} \langle\varphi(x), \theta'_{h,a}\rangle\right| \leq \sup_{x\in\mathcal{S}}\max_{a\in\mathcal{A}}\left|\langle\varphi(x), \theta_{h,a} - \theta'_{h,a}\rangle\right| \\
&\leq R\max_a\left\|\theta_{h,a} - \theta'_{h,a}\right\|.
\end{aligned}
$$

Thus, the $\varepsilon/2$-covering number of the set $\Theta = \{(\theta_a)_{a\in\mathcal{A}} : \theta_a \in \mathbb{R}^d, \ \sup_{a\in\mathcal{A}} \|\theta_a\|_2 \leq \beta\}$ is bounded by $(1 + 4\beta/\varepsilon)^{Ad}$, which follows from the standard fact that the $\varepsilon$-covering number of a ball in $\mathbb{R}^d$ with radius $c > 0$ in terms of the $\ell_2$ distance is bounded by $(1 + 2c/\varepsilon)^d$. Thus, we have that for any $V_h \in \mathcal{V}_h$, there exists a $\widetilde{V}_h$ parameterized by $\widetilde{\theta}_h$ in the $\varepsilon/2$-covering of $\Theta_h$ such that,

$$
\|V_h - \widetilde{V}_h\|_\infty \leq R\varepsilon/2.
$$

By rescaling of the covering numbers, we can see that the logarithm of the $\varepsilon$-covering number of $\mathcal{V}_h$ can be bounded by

$$
\log(\mathcal{N}(\mathcal{V}_h, \varepsilon)) \leq \log(\mathcal{N}(\Theta_h, \varepsilon/(2R))) \leq dA\log(1 + 4\beta R/\varepsilon),
$$

giving the result.

## Footnotes

[7]To see this, consider $\tilde{p}$ and $\tilde{p}'$ satisfying the constraints, which differ only in $x$ where $\tilde{p}(x) = p_0$ and $\tilde{p}'(x) = 0$. Then, nontrivial convex combinations of $\tilde{p}, \tilde{p}'$ no longer satisfy the constraints.