[Reviews · NeurIPS 2020]

Review 1

Summary and Contributions: This theoretical paper uses duality to show that model optimistic algorithms have a much more practical value optimistic analogue. While calculating the value optimistic exploration bonuses turns out to be difficult as well, a much easier upper bound can be used to derive an approximation whose regret bound is within a constant of the original algorithm.

Strengths: Model optimistic and value optimistic algorithms are considered to be fundamentally different. By showing that one can approximately reduce one to the other, this paper helps unify these approaches. It is possible that these results could help us design algorithms that are of practical use and retain some theoretical guarantees, or at least allow us to advance the field without further complicating analysis.

Weaknesses: The main paper and appendix total 34 pages. It's not clear that this makes sense for a conference paper. It would have been better to focus on the tabular case in this paper and defer results for factored linear MDPs to a followup. That way the paper could have been restricted to a more reasonable length, and essential results could have been included in the main body of the paper. The contribution of this paper is purely theoretical. Neither the original model optimistic algorithms, nor the easier to implement value optimistic duals are mature enough to be used in a real application. While a step in the right direction, I expect the audience of this paper to be limited to a handful of authors who may build upon these results. The motivation provided in the paper is that model optimistic algorithms are easier to analyze, while value optimistic algorithms are easier to implement. This may be more a reflection of the authors' background, rather than a universal truth. Some authors find thinking in terms of value functions more intuitive. Nevertheless the paper should prove useful to authors from both backgrounds.

Correctness: While the claims in the main body of the paper seem reasonable, I did not carefully go through the proofs in the 23 page appendix, so I cannot confidently confirm correctness.

Clarity: The paper is well written and organized.

Relation to Prior Work: The paper does a good job relating its results to the state of the art, as well as rederive some recent results using the proposed framework.

Reproducibility: Yes

Additional Feedback: Reference 42 is missing venue and year. Post-rebuttal: Thank you for your responses. After some discussion I do understand R2's concern about the applicability of the duality result, though I find it to be of interest in and of itself. It would be great if you could provide some suggestions on how the duality result could be used to improve the theory or practice of exploration in RL.


Review 2

Summary and Contributions: This paper uses Lagrangian duality to show that every existing model-optimistic algorithm has an equivalent representation as a value-optimistic algorithm, which is computationally efficient. Based on this connection, this paper derives a class of algorithms that are modifications of existing model-optimistic algorithms but can be implemented by DP. This paper further studies the linear function approximation setting and obtains matching bounds as in previous works.

Strengths: 1. The duality between model-optimistic algorithms and value-optimistic algorithms is an interesting finding. 2. This paper is very well-written.

Weaknesses: While I like the duality result, I find this paper is not substantial enough that merits acceptance. This paper shows a class of model-optimistic algorithms can be implemented efficiently (with minor modifications). However, none of state-of-the-art algorithms is model-optimistic algorithms. This is somehow inherent with this class of algorithms because the transition model scales with $S^2$ but the optimal bounds scale linearly in $S$ via value-optimistic algorithms. Value-optimisic algorithms are not only more computationally efficient but also more statistically efficient. So making model-optimistic algorithms more efficient is not a very significant result. I would be happy to recommend acceptance if this paper contains any of the following results either 1) A substantially simpler analysis of existing SOTA algorithms, like UCB-VI-Bernstein, EULER (Zanette et al. 2019), ORLC (Dann et al. 2019), UCB-Q-Bernstein the framework in this paper. 2) A new model-optimistic-inspired algorithm which achieves SOTA sample complexity ($H\sqrt{SAT}$). Tighter Problem-Dependent Regret Bounds in Reinforcement Learning without Domain Knowledge using Value Function Bounds Andrea Zanette, Emma Brunskill Policy Certificates: Towards Accountable Reinforcement Learning Christoph Dann, Lihong Li, Wei Wei, Emma Brunskill

Correctness: Yes.

Clarity: Yes. This paper explains this its idea very well.

Relation to Prior Work: Yes.

Reproducibility: Yes

Additional Feedback: I've read the rebuttal and decided to keep my score. The response does not address my concern about lack of substantial application of the duality result. --------------------------------------------------------------- Minor comments: 1. line 86-86 missing summations. 2. UCB-VI only matches the lower bound in the regime $SA \ge H$ and $T$ is sufficiently large. EULER and ORLC match the lower bound and do not require $SA \ge H$, though they still require $T$ sufficiently large. 3. There are some typos in references, e.g., [42] only has a page number.


Review 3

Summary and Contributions: The paper proves that any model-optimistic algorithm can be transformed to an equivalent value-optimistic algorithm. This enables cleaner theoretical analysis of an optimistic algorithm through its model-optimistic formulation while allowing easier implementation through its value-optimistic formulation.

Strengths: While I could not check proofs of all claims, they seems to be valid. The result that any model-optimistic algorithm can be expresses as both value- and model-optimistic algorithms is, I think, novel. Also this result is very significant because it enables cleaner theoretical analysis of an optimistic algorithm while allowing its easier implementation by dynamic programming.

Weaknesses: I do not find any particular weakness of the paper.

Correctness: I did not check proofs in the appendix, but Proposition 1, which is the main result showing an equivalence of model-optimistic algorithms to value-optimistic algorithms, looks correct (as far as I can tell from its proof sketch in the main paper).

Clarity: Yes, it is clearly written and easy to follow.

Relation to Prior Work: Yes.

Reproducibility: Yes

Additional Feedback: I like the paper very much. I have two question to the authors. Regarding Table 1, is it possible to find an optimal D such that the regret bound is tightest? If no, what makes find an optimal D difficult? If yes, what does it look like? Is there a hope that model-optimistic algorithms can archive a minimax regret bound with such an optimal D? Regarding Theorem 4, is it possible to get something similar for PAC, or uniform-PAC bounds (https://arxiv.org/pdf/1703.07710.pdf)? It would make the paper stronger, I think. ----- After Reading Author Feedback ----- I read the author feedback and other reviews. The author feedback answered my questions very well. I agree with other reviewers that the paper would be stronger if it has either a substantially simpler analysis of existing SOTA algorithms or a model-optimistic algorithm with a SOTA sample complexity, as an application of the duality result. However, I think the duality result itself is interesting and has potential to be used in other papers to improve algorithms. Therefore, I keep my score as is.


Review 4

Summary and Contributions: The paper proposes a new framework for designing and analyzing optimistic RL algorithms in episodic tabular MDPs. The paper also proposes a few other technical proofs that might be of independent interest. --- post rebuttal --- I am not an expert in this area and I do not feel adequate to judge this work. I have deferred judgement to the remaining reviewers.

Strengths: The paper is very clearly written and organized. I am not an expert in this area and therefore not able to judge the significant and novelty of the contribution.

Weaknesses: The paper is, if I am not mistaken, entirely theoretically. It would be insightful to have experiments, even on toy environments where the optimal policies can be computed, to understand to what extent the theoretical analysis transfer to experiments.

Correctness: I did not get the chance to go over the proof in the appendix.

Clarity: Yes. The paper is very well written and organized.

Relation to Prior Work: I would have liked to see more discussions on the connections between the paper, its theoretical results and some of the more experimental papers, especially RL method that uses optimism for exploration.

Reproducibility: Yes

Additional Feedback:

[Author Response · NeurIPS 2020]

¹ We thank all reviewers for their careful reading of our paper and their constructive feedback. We respond to each
² reviewer separately below.

³ **Reviewer #1**    Thank you for expressing your appreciation of our results and your thoughtful comments! Regarding
⁴ your remarks:

- ⁵ While the paper is indeed somewhat lengthy, its length is far from being unusual for theoretical papers at
  ⁶ NeurIPS, especially in this topic. For instance, we refer to some relevant papers from NeurIPS 2019: "Non-
  ⁷ Asymptotic Gap-Dependent Regret Bounds for Tabular MDPs" (54 pages), "Exploration Bonus for Regret
  ⁸ Minimization in Discrete and Continuous Average Reward MDPs" (34 pages), "Tight Regret Bounds for
  ⁹ Model-Based Reinforcement Learning with Greedy Policies" (54 pages).

- ¹⁰ We agree that theoretically sound optimistic RL algorithms are still not quite mature enough for real-world
  ¹¹ applications, but we would like to point out that there is steady progress in this field of research, with several
  ¹² new results being published at NeurIPS, ICML and COLT each year. Our work will be of interest to members
  ¹³ of the community working on theoretical RL along with those working to bridge this gap between theory and
  ¹⁴ practice, and so we believe the audience of our work will extend beyond "a handful of authors".

- ¹⁵ We absolutely agree that our assertion about the model-optimistic framework yielding a simpler analysis is
  ¹⁶ subjective, and we hope that our writing didn't suggest otherwise. We will make this clearer for the final
  ¹⁷ version. In any case, we believe that many researchers may share our views about the simplicity of the analysis
  ¹⁸ of model-based approaches and will find our results insightful.

¹⁹ **Reviewer #2**    We absolutely agree that addressing the gap between the best available bounds for model-optimistic and
²⁰ value-optimistic algorithms is an important open challenge, and that our paper did not manage to close this gap. In
²¹ the present paper, we dispel the commonly held belief that model-optimistic methods would be difficult to implement,
²² which at the very least suggests that such algorithms may be more powerful than they are commonly thought to be. In
²³ fact, we think that there is still no sufficient evidence for the claim that "value-optimisic algorithms [would be] more
²⁴ statistically efficient" than model-optimistic ones[1], and believe that the primal-dual view we introduce in this paper may
²⁵ prove beneficial for making progress towards addressing this important question. For instance, our primal view could
²⁶ enable constructing more sophisticated (non-local) confidence sets for the transition functions that could lead to tighter
²⁷ performance guarantees. In Section 5 of our paper, we demonstrate that this is indeed the case for linear MDPs where a
²⁸ model-based perspective leads to state of the art algorithms. In the tabular setting, as the reviewer points out, closing
²⁹ the gap is more difficult, and beyond the scope of the present (already lengthy) paper. However, we believe that the
³⁰ results presented in our work will serve as an important stepping stone towards developing such extensions. We will
³¹ add further discussion of this to the final version of the paper.

³² **Reviewer #3**    Thank you for your positive evaluation of our paper! Regarding your questions:

- ³³ Finding an "optimal $D$" is indeed a very interesting question, but also a rather complex one. For a statistically
  ³⁴ valid analysis, one has to jointly pick a divergence $D$ and a confidence width $\epsilon$, in a way that the primal
  ³⁵ confidence intervals remain valid and the exploration bonuses (defined in terms of the conjugate $D_*$) are
  ³⁶ small. The interdependence of these factors makes it difficult to reverse-engineer a divergence from confidence
  ³⁷ bounds used by existing value-optimistic algorithms, and so far, we have not been able to derive tighter bounds
  ³⁸ following this (otherwise very tempting) approach. We believe that achieving minimax-optimal regret bounds
  ³⁹ may necessitate using more sophisticated confidence bounds in place of the local ones we analyze in this paper,
  ⁴⁰ but we leave this to future work.

- ⁴¹ It is definitely possible to derive (uniform) PAC bounds using the techniques developed in our paper, since
  ⁴² PAC-MDP algorithms also rely on the same notion of optimism as used for proving regret bounds. The key
  ⁴³ technical step in both PAC and regret analyses is bounding the instantaneous regret $\Delta_{1,t}$ in each episode, and
  ⁴⁴ then bounding $\sum_t \Delta_{1,t}$ to obtain a bound on the regret, or $\sum_t \mathbb{I}_{\{\Delta_{1,t} > \varepsilon\}}$ to obtain a PAC bound. Thus, one
  ⁴⁵ can also derive PAC bounds by following the steps in Theorem 4. We focused on regret bounds in the present
  ⁴⁶ paper since this framework is the most well-studied, but we will point out the possibility of providing PAC
  ⁴⁷ bounds in the final version of the paper—thank you for suggesting this extension!

⁴⁸ **Reviewer #4**    Thank you for your review. In particular, thanks for the suggestion of adding more discussion about the
⁴⁹ relationship between this work and empirically successful work. We will do this in the final version.

## Footnotes

[1]This view is currently only supported by comparing *upper bounds* without a separating lower bound.


[Meta-Review · NeurIPS 2020]

The reviewers are in agreement that this is interesting and well-presented work. The main concern was about the extent to which the results will help us derive SOTA algorithms in the future. I find the contribution reasonable without this and hope the community will figure out how/if these results are useful. Please do take the reviewers minor suggestions into consideration when preparing a final version.